# Rectifying artificial nanochannels with multiple interconvertible permeability states

Ruocan Qian [1,2,3,4,7] ✉, Mansha Wu[1,2,3,4,7], Zhenglin Yang[5], Yuting Wu [5], Weijie Guo[6], Zerui Zhou[1,2,3,4], Xiaoyuan Wang[1,2,3,4], Dawei Li[1,2,3,4] & Yi Lu [5,6] ✉

Transmembrane channels play a vital role in regulating the permeation process, and have inspired recent development of biomimetic channels. Herein, we report a class of artificial biomimetic nanochannels based on DNAzyme-functionalized glass nanopipettes to realize delicate control of channel permeability, whereby the surface wettability and charge can be tuned by metal ions and DNAzyme-substrates, allowing reversible conversion between different permeability states. We demonstrate that the nanochannels can be reversibly switched between four different permeability states showing distinct permeability to various functional molecules. By embedding the artificial nanochannels into the plasma membrane of single living cells, we achieve selective transport of dye molecules across the cell membrane. Finally, we report on the advanced functions including gene silencing of miR-21 in single cancer cells and selective transport of $Ca^{2+}$ into single PC-12 cells. In this work, we provide a versatile tool for the design of rectifying artificial nanochannels with on-demand functions.

Transmembrane channels play a crucial role in myriad biological events, including regulating membrane potentials, virus entry into host cells, intercellular communications, electrical signaling, and neuronal dysfunction, via precisely controlled directional flow of ions[1-4]. The permeability of transmembrane channels can change swiftly under external stimuli, thus enabling the stimuli-responsive control of the transport properties of biological transmembrane channels[5-8]. To better understand the mechanism of transmembrane channels-related bio-pathways, designing artificial channels that mimic the selective permeation function of biological channels has been reported and has shown huge promise for potential applications in biomedical research[9-11].

$Ca^{2+}$ ions are known to contribute to many biological processes and physiological activities of living cells, and are of great significance in maintaining the homeostasis of the body[12]. As a ubiquitous second messenger, $Ca^{2+}$ ions can control various cellular biological functions, including energy metabolism, cellular differentiation, proliferation, survival, and apoptosis[13]. Therefore, $Ca^{2+}$ channels play a vital role in the cell by allowing for the selective import and export of $Ca^{2+}$ ions to modulate various cellular functions. This process is known as $Ca^{2+}$ signaling, which is reported to be involved in various diseases, including cancer, autoimmune diseases, and virus infection[14]. To construct biomimetic ion channels, various functional materials, such as hierarchically organized polymers, DNA nanopores, G-quadruplexes, metal-organic cuboctahedra, and solid-state nanopores have been used[15-20]. These bio-inspired artificial nanochannels are designed to mimic selective transmembrane transport. While much progress has been made in this area, new types of materials are needed to make the biomimetic transmembrane channels less intricate to carry out and easier to be functionalized so that they can be employed for a wider

[1]Key Laboratory for Advanced Materials, East China University of Science and Technology, Shanghai 200237, P. R. China. [2]Feringa Nobel Prize Scientist Joint Research Center, Joint International Laboratory for Precision Chemistry, East China University of Science and Technology, Shanghai 200237, P. R. China. [3]Frontiers Science Center for Materiobiology & Dynamic Chemistry, East China University of Science and Technology, Shanghai 200237, P. R. China. [4]School of Chemistry and Molecular Engineering, East China University of Science and Technology, Shanghai 200237, P. R. China. [5]Department of Chemistry, University of Texas at Austin, Austin, TX 78712, USA. [6]Department of Molecular Biosciences, University of Texas at Austin, Austin, TX 78712, USA. [7]These authors contributed equally: Ruocan Qian, Mansha Wu. ✉e-mail: ruocanqian@ecust.edu.cn; yi.lu@utexas.edu

application in biomedical research. Toward this goal, glass capillary-based nanopipettes have recently been reported to build biomimetic nanochannels because of their submicron-sized conical tip, easy modification, and convenient insertion into the cell membrane[21–24]. To transform such a system into a biomimetic ion channel that is responsive to different ions, the inner surface of these glass nano-pipettes needs to be modified with various functional groups[25,26]. For example, Xiong et al. reported a polyimidazolium brush (PimB)-con-fined fluidic channel, which served as a polyelectrolyte-confined fluidic memristor (PFM) to perform various neuromorphic functions based on the spatial confinement and fluidic-based ion redistribution dynamics of PFMs[27]. Besides, i-motif DNA strands have also been used for nanopipette functionalization[28]. The i-motif DNA structure could be destroyed after the addition of glutathione (GSH) due to the strong interaction of Ag-S bond, which results in the recovery of surface charge at the inner wall of the nanopipette, causing the subsequent change of ion current rectification. However, most of the existing studies focus on single stimulus-triggered changes between two per-meability states[29–31]. Indeed, biological transmembrane channels often respond to more than one stimulus to exhibit multiple permeability states[32,33]. Therefore, despite recent progress made in this field, it remains challenging to design a simple strategy to realize reversible and selective regulation of channel surface properties between mul-tiple distinct permeability states under various external stimuli.

In this work, we report the design of a smart and convenient strategy to build artificial biomimetic nanochannels based on glass nanopipettes to realize delicate control of channel permeability using RNA-cleaving DNAzymes for inner surface modification. DNAzymes are a class of DNA molecules that display enzymatic activity, with a specific metal ion as a cofactor[34–37]. Since their first discovery in 1994, DNAzymes have been widely applied in metal ion sensing in solutions or in living systems, as the susceptibility of DNAzymes to metal-dependent cleavage can effectively induce the breakage of substrate ribonucleotides[38,39]. Here, we employ metal ion-specific RNA-cleaving DNAzymes and their substrates as functional molecules for the mod-ification of glass nanopipettes to build artificial nanochannels. The corresponding DNA double-chain switches enable the manipulation of wettability and charge at the inner surface under external stimuli (metal ions or DNAzyme-substrates). We have demonstrated the reversible switching of the nanopipettes between four different per-meability states (state 1: hydrophobization, negative rectification; state 2: hydrophilization, negative rectification; state 3: hydrophobization, positive rectification; state 4: hydrophilization, positive rectification), which exhibited distinct permeability to different functional mole-cules. Using the truncated tip of the nanopipettes, we have embedded the artificial nanochannels into the membrane of single living cells, which enabled selective transport of bioactive species across the cell membrane. Our work provides a versatile tool for the design of recti-fying artificial nanochannels with on-demand functions, including controlled gene silencing of miR-21 in single cancer cells and selective transport of $Ca^{2+}$ into single PC-12 cells.

## Results

### Design of the artificial nanochannels based on DNAzyme-functionalized nanopipettes

Biological channels share a common feature of containing narrow pore channels with selective filter regions to achieve fast and selective transport of various bioactive species in and out of cells[40]. The selec-tive transport mechanism of biological channels has been considered as an efficient regulator to control the permeation of ions and molecules[41]. In some particular conditions, the inner surface property of the channels can be reversed to allow the passing through of certain ions or molecules. A typical example is the Orai calcium channels. As shown in Fig. 1a, in the closed state of Orai channel, the permeability of $Ca^{2+}$ is blocked. Upon activation, the exposure of alkaline amino acid-

rich region causes strong attraction of anions, which enables $Ca^{2+}$ permeation[42].

In order to mimic biological channels to allow precisely controlled directional flows of different species, we propose to modify the inner wall of the nanopipettes with two DNAzymes (Fig. 1b). The DNAzyme 1 is a $Mg^{2+}$ specific RNA-cleaving DNAzyme consisting of an enzyme strand whose 5′-end is conjugated with an amino group. The corre-sponding Substrate 1 is tagged with a cholesterol tail at 5′-end, so the wettability of the inner wall of nanopipette could be tuned by cutting the cholesterol tail[43]. The DNAzyme 2 is a $Zn^{2+}$ specific RNA-cleaving DNAzyme, consisting of an enzyme strand with an amino tail at 3′-end, and its Substrate 2 with a carboxyl tail at 5′-end. Once these two pairs of DNAzymes and their corresponding substrates are hybridized and then covalently linked to the inner surface of glass nanopipettes (the pretreatment process of nanopipettes has been described in the Experimental section)[44], the cholesterol group of Substate 1 and the carboxyl group of Substrate 2 are exposed, resulting in a hydro-phobic and negatively charged inner surface.

To further realize the control of inner surface properties for reversible conversion between different permeability states of nano-channels, specific cofactor metal ions or substrates of the DNAzymes are added into the nanochannels to either cleave the substrate into products or replenish the cleaved substrates. As shown in Fig. 1c, d, for example, the exposure of cholesterol and carboxyl groups leads to a hydrophobic and negatively charged state (state 1), which can be transferred into a negative but hydrophilic state (state 2) due to the cleavage of substrate 1 (sub-1), resulting in the leaving of cholesterol tail after the addition of $Mg^{2+}$. In contrast, the state of the nanochannel can also be transferred from state 2 to state 1 by adding sub-1. Similarly, the addition of $Zn^{2+}$ enables the cleavage of substrate 2 (sub-2), leading to the exposure of amino groups. Thus the inner surface property is transferred from a hydrophilic and negative state (state 2) to a hydrophilic but positive state (state 4). The addition of sub-2 allows the reversible transfer from state 4 to state 2. Therefore, the permeability of the artificial nanochannel can be reversibly switched between the four different states: state 1 (hydrophobization, negative rectification), state 2 (hydrophilization, negative rectification), state 3 (hydro-phobization, positive rectification), and state 4 (hydrophilization, negative rectification) (Supplementary Table 1). The nanopipettes are filled with electrolyte solution and inserted with an Ag/AgCl electrode for the characterization of hydrophilicity and rectification inside the artificial nanochannels. The melting temperatures of the DNAzymes are ~60 °C, much higher than 37 °C for physiological conditions. Therefore, the hybridized DNA chains are stable at 37 °C. In contrast, the melting temperatures of the halves of the cleaved substrates are lower (~20 °C) than the physiological temperature so that the sub-strates can be cut upon metal treatment at 37 °C (Supplementary Table 2). Using DNAzyme-functionalized nanopipettes as a proof of concept, we aim to achieve delicate control of the permeability properties of the nanochannels under the treatment of metal ions and DNAzyme-substrates.

Before functionalization, bare glass nanopipettes with a typical conical shape were prepared by pulling quartz capillaries for the fol-lowing preparation of rectifying artificial nanochannels (Supplemen-tary Table 3). The side view and top view of the nanopipettes before and after DNAzyme modification are shown in Fig. 1e. To find out the best modification strategy, we used two different methods to achieve the inner surface modification. The first method was to induce alde-hyde groups on the inner glass surface to bind with amino group-labeled DNAzyme ($NH_2$-DNA, Supplementary Fig. 1). The second method was to induce Au cladding on the inner glass surface to bind with thiol-labeled DNAzyme (SH-DNA, Supplementary Fig. 2). From ion current rectification (ICR) curves, we can see that bare nanopipettes showed a slightly negative rectified ionic current owing to the dis-sociation of silanol groups at the inner surface (control). For the first

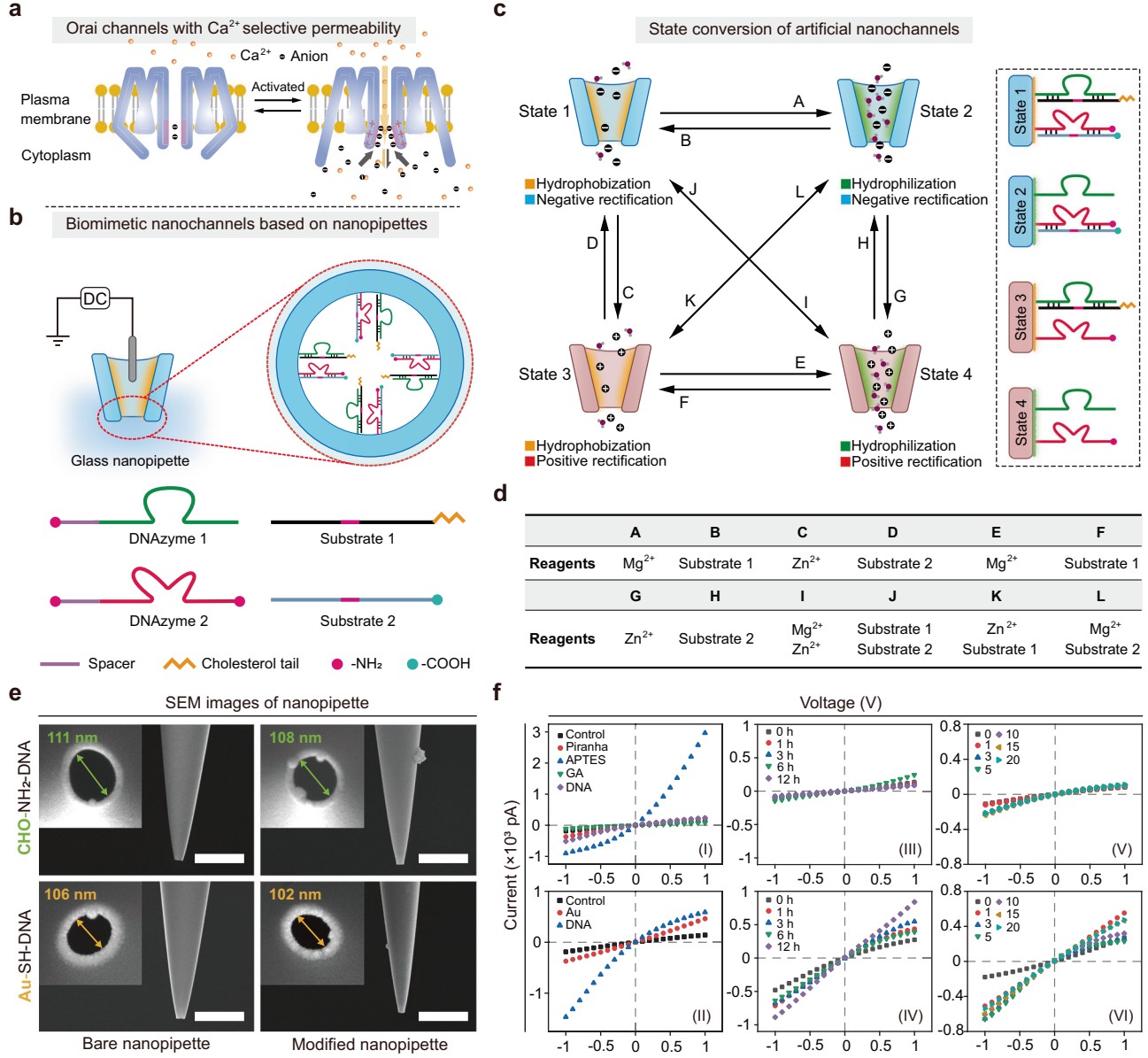

**Fig. 1 | Fabrication and characterization of the DNAzyme-functionalized glass nanopipettes. a** Orai channels with Ca²⁺ selective permeability in nature. **b** Biomimetic nanochannels constructed by DNAzyme-functionalized nanopipettes. DNAzyme 1: Mg²⁺ specific DNAzyme. Substrate 1: substrate strand of DNAzyme 1 with a cholesterol tail. DNAzyme 2: Zn²⁺ specific DNAzyme with an amino tail. Substrate 2: substrate strand of DNAzyme 2 with a carboxyl tail. **c** Left: Inter-convertible control of wettability and charge of the inner wall via metal ions and DNAzyme-substrates between four different permeability states. Right: four different states corresponding to different hydrophilicity and rectification features on the surface of the inner wall. State 1: hydrophobization, negative rectification. State 2: hydrophobization, negative rectification. State 3: hydrophobization, positive rectification. State 4: hydrophilization, positive rectification. **d** Reagents needed for state conversion of artificial nanochannels corresponding to A to L in **c**. **e** SEM images of nanopipettes before and after DNAzyme modification. Up: nanopipettes functionalized with DNAzyme via aldehyde modification to bind with amino group-labeled DNAzyme. Down: nanopipettes functionalized with DNAzyme via Au cladding modification to bind with thiol-labeled DNAzyme. Experiments were repeated three times independently with similar results. **f** Current-voltage (*I–V*) curves measured in a single nanopipette showing the modification procedure based on linkage between (I) aldehyde-modified inner surface and amino group-labeled DNAzyme and (II) Au cladding modified inner surface and thiol-labeled DNAzyme. (III) & (IV) *I–V* curves showing the stability of a DNA-functionalized nanopipette after different times corresponding to (I) and (II). (V) and (VI) *I–V* curves measured in a DNA-functionalized nanopipette upon different rounds of washing corresponding to (I) and (II). Scale bar: 500 nm. Source data are provided as a Source Data file.

method, the nanopipette was treated with piranha solution to obtain hydroxyl groups on the inner surface, and then treated with 3-aminopropyl-triethoxylsilane (APTES) to induce amino groups, which led to a positive rectified ionic current. Afterward, the nanopipette was treated with glutaraldehyde (GA) to induce electrically neutral aldehyde groups, resulting in an almost linear *I–V* curve (no rectification). After DNAzyme modification, a significant negative ICR was observed, indicating successful modification of DNAzyme (CHO-NH₂-DNA,

Fig. 1f, I). For the second method, the inner wall of the nanopipette was first coated with a thin layer of gold film. From UV-vis characterization (Supplementary Fig. 3), the absorption peak of AuCl₄⁻ inside the nanopipette was noticeably decreased, indicating successful deposition of Au[28]. Afterwards, the FAM-labeled SH-DNA was injected into the gold-coated nanopipette and thiolated for 8 h. The DNAzyme solution before and after the reaction was tested by fluorescence (FL) spectra, and the FL intensity of FAM was obviously decreased, indicating

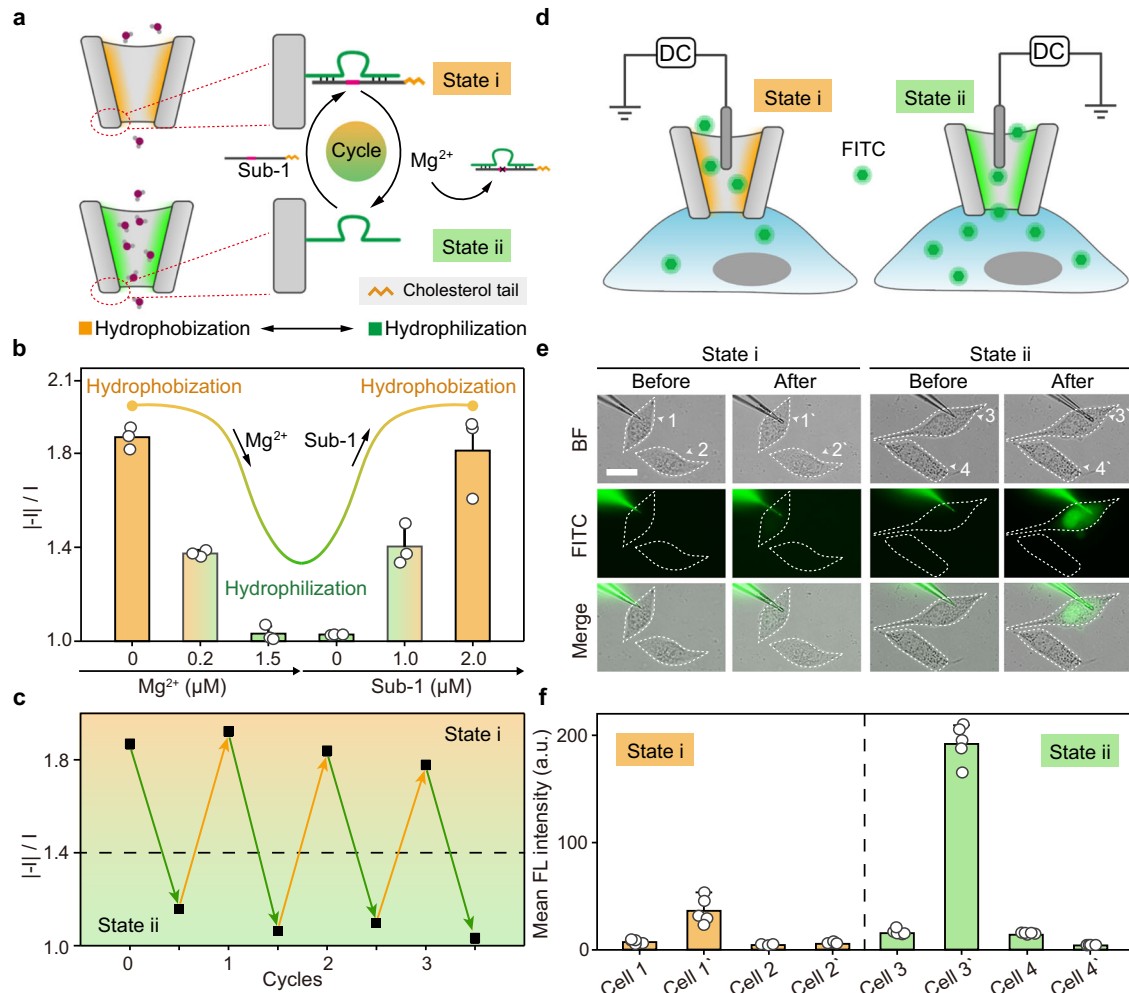

**Fig. 2 | Reversible wettability control in a nanopipette. a** Design of a DNAzyme-based switch (DNAzyme 1 and substrate 1) for inner surface wettability control in a nanopipette (wettability-reversal nanochannel). State i: hydrophobic. State ii: Hydrophilic. **b** Current rectification ratio measured in a wettability-reversal nanochannel upon the treatment of different concentrations of $Mg^{2+}$ and substrate 1 (sub-1). **c** The reversibility of the wettability-reversal nanochannel. Downward arrow: $Mg^{2+}$ treatment. Upward arrow: sub-1 treatment. **d** Schematic illustration showing transport of FITC in a wettability-reversal nanochannel under a DC voltage

into single living HeLa cells. **e** Microscopic cell images showing the outflow of FITC into single living HeLa cells. BF bight-field. FITC green fluorescence. Merge: mixed green & bright-field channel. Scale bar: 20 μm. **f** FL intensity comparison histogram of FITC in **e**. Error bars in **b** represent the standard deviation of three independent experimental repeats, and the measure of the center represents their corresponding mean value. Error bars in **f** represent the standard deviation of five independent experimental repeats, and the measure of the center represents their corresponding mean value. Source data are provided as a Source Data file.

successful modification of DNAzyme (Supplementary Fig. 4). From the ICR curves, Au-coated nanopipette showed a more negative rectified ionic current due to the strong absorption of $Cl^-$ on the surface of Au[45]. After DNAzyme modification, the ICR curve became more negative, which further confirmed a successful modification of DNAzyme (Au-SH-DNA, Fig. 1f, II)[46]. The stability of DNAzyme-functionalized nanopipettes prepared by the above two methods was compared by measuring *I–V* curves after different times (Fig. 1f, III, IV) or different rounds of washing (Fig. 1f, V, VI), and we can see that the linkage between -CHO and -$NH_2$ was more stable. Therefore, the first method was selected for all the following studies, with optimized length and concentration of the DNAzyme assisted by spacers (Supplementary Figs. 5–7).

## Reversible wettability control in a nanopipette

In the initial stages of constructing the DNAzyme-based double control switches (Fig. 2a), we prepared nanochannels functionalized with a single pair of double chains composed of DNAzyme 1 (a $Mg^{2+}$-specific DNAzyme) and its substrate 1 (sub-1, tagged with cholesterol tail) to mimic the natural nanochannels that are switchable between dewetted (hydrophobic) and wetted states (hydrophilic). Upon hybridization

between DNAzyme 1 and sub-1, the inner surface of the nanochannel was modified by the hybridized DNA (dsDNA) initially (state i). When $Mg^{2+}$ was added, the sub-1 strand was cleaved, and the nanochannel was then switched from state i to state ii, with its inner surface modification changed to single-stranded DNA (ssDNA). Due to the negative charge of DNA backbone, the resulting dsDNA induced a more negatively charged surface with higher ICR ratio in state i than the ssDNA cleaved products in state ii (Supplementary Fig. 8a), because the conformation of the dsDNA exposed the negatively charged backbone to the surface more fully than the ssDNA. As shown in Fig. 2b, the addition of $Mg^{2+}$ led to the cleavage of cholesterol tagged sub-1, thus the inner surface of the nanochannel changed from state i containing dsDNA to state ii containing ssDNA. Due to the negative charge of DNA backbone, the conformation of the dsDNA exposed the negatively charged backbone to the surface more fully than the ssDNA, the negatively charged $[Fe(CN)_6]^{3-}/[Fe(CN)_6]^{4-}$ would face greater resistance in approaching to the dsDNA modified surface (state i) than to the ssDNA modified surface (state ii). Therefore, state i showed a higher charge-transfer resistance ($R_{ct}$) compared with state ii (Supplementary Fig. 8b)[47]. In addition, the zeta potential of the dsDNA

modified surface (state i) was −58.93 mV, which was more negative than that of state ii (−49.89 mV), indicating a more negative surface charge at state i (Supplementary Fig. 8b). Besides, static contact angle experiments showed that the contact angle decreased from $(46.43 \pm 1.21°)$ to $(26.39 \pm 1.97°)$ after the addition of $Mg^{2+}$ (Supplementary Fig. 8c), indicating that the cleavage of cholesterol tagged sub-1 induced a more hydrophilic surface at state ii than that at state i. Thus, the addition of $Mg^{2+}$ transformed the surface wettability from hydrophobic to hydrophilic The DNAzyme-controlled substrate cleavage in the presence of $Mg^{2+}$ was confirmed by gel electrophoresis (Supplementary Fig. 9). The above results demonstrated that the nanochannel could be reversibly switched between state i and state ii. Furthermore, the state variation of the nanochannel was reversible for more than three cycles (Fig. 2c), which illustrated the capability of DNAzyme-based nanochannel to achieve accurate and reversible control of inner surface wettability. Two control experiments were performed to check the availability and specificity of the DNAzyme-based switching between hydrophobic and hydrophilic (Supplementary Fig. 10). First, inactive DNAzyme 1 (chain-7) and sub-1 were used to modify the nanochannel, and no change of rectification was observed upon the addition of $Mg^{2+}$ and sub-1 (Supplementary Fig. 10a). Second, different metal ions ($Fe^{2+}$, $K^+$, $Ca^{2+}$, $Zn^{2+}$, $Na^+$, $Ba^{2+}$, $Ag^+$, 2 μM each) were added as the input signal, and the results showed a negligible change of rectification (Supplementary Fig. 10b). The above results demonstrated the DNAzyme-based wettability switching inside the functionalized nanochannel. To mimic controlled directional flow in natural transmembrane channels, a lipophilic dye, FITC, was selected to test the permeability of the functionalized nanochannel into a single living cell (Fig. 2d). Using HeLa cells as the example, the tip of the nanopipette was inserted into the cell membrane (Supplementary Fig. 11), while a DC voltage was applied to inject FITC into single HeLa cells, as the FITC molecules were contained inside the nanopipettes under different states. As shown in Fig. 2e, the FL of intracellular FITC after injection was significantly higher in cell 3', indicating an increased lipid permeability in state ii. Compared with state i, the nanochannel in state ii was less attractive to FITC molecules, resulting in significantly increased transport of FITC into cells (Fig. 2f).

## Reversible charge control in a nanopipette

Building on our previous work, we developed a charge control switch based on a similar DNA double-chain structure. Nanochannels functionalized with a single pair of double chains composed of DNAzyme 2 and substrate 2 (sub-2) were prepared to mimic the natural nanochannels, which are switchable between negative rectification and positive rectification. As shown in Fig. 3a, DNAzyme 2, a $Zn^{2+}$-specific DNAzyme with an amino-terminal, and its substrate, sub-2 (with a carboxyl-terminal), were used to build a DNA double-chain control switch for inner surface charge regulation. Upon double-chain hybridization, the nanochannel presented a negative rectification (state iii) initially due to the negative DNA frameworks and the carboxyl groups at the end of sub-2. When $Zn^{2+}$ was added as the input signal, sub-2 strand could be cleaved from the ribonucleotide cleavage site, causing immediate falling out of the carboxyl groups, and the nanochannel was then switched from state iii to state iv (positive rectification), as the amino groups at the end of DNAzyme 2 were exposed (Supplementary Fig. 12a). As shown in Supplementary Fig. 12b, state iii showed a higher $R_{ct}$ than that of state iv, indicating a more negatively charged surface in state iii. Static contact angle experiments showed similar contact angles of state iii ($27.27 \pm 0.98°$) and state iv ($24.52 \pm 0.65°$) (Supplementary Fig. 12c). To validate DNAzyme-controlled substrate cleavage, the DNAzyme 2/sub-2 mixture was treated with $Zn^{2+}$ and analyzed by gel electrophoresis, which confirmed the cleavage of the substrate in the presence of $Zn^{2+}$ (Supplementary Fig. 13). As shown in Fig. 3b, in the presence of $Zn^{2+}$, the ICR ratio decreased with the increasing $Zn^{2+}$ concentration. In contrast, upon the following supplement of sub-2,

the ICR ratio could be recovered. The above results demonstrated that the nanochannel could be reversibly switched between state iii and state iv. Also, the state variation of the nanochannel was reversible for more than three cycles (Fig. 3c), which illustrated the capability of DNAzyme-based nanochannel to achieve accurate and reversible control of inner surface charge. Two control experiments were performed to check the availability and specificity of the DNAzyme-based switching between negative and positive rectification (Supplementary Fig. 14). First, inactive DNAzyme 2 (chain-8) and sub-2 were used to modify the nanochannel, and no change of rectification was observed upon the addition of $Zn^{2+}$ and sub-2 (Supplementary Fig. 14a). Second, different metal ions ($Ag^+$, $Mg^{2+}$, $K^+$, $Na^+$, $Ba^{2+}$, $Fe^{2+}$, $Ca^{2+}$, 2 μM each) were added as the input signal, and the results showed a negligible change of rectification (Supplementary Fig. 14b). The above results demonstrated the availability of DNAzyme-based charge switching inside the functionalized nanochannel. To mimic controlled directional flow in natural transmembrane channels, methylene blue (MB, a commonly used cationic dye for biological staining with red fluorescence, Supplementary Fig. 15) was selected to test the permeability of the functionalized nanochannel (Fig. 3d). A DC voltage was applied to inject MB into single HeLa cells, as the MB molecules were contained inside the nanopipettes under different states. As shown in Fig. 3e, the FL of intracellular MB after injection was significantly higher in cell 3', indicating a significantly enhanced cation permeability in state iv. Compared with state iii, the nanochannel in state iv was less attractive to MB molecules, resulting in significantly increased transport of MB into cells (Fig. 3f).

## Rectifying artificial nanochannels with four interconvertible permeability states

In order to achieve more delicate control for the regulation of wettability and charge, the inner wall of the nanopipettes was modified with two pairs of double chains, DNAzyme 1/sub-1 and DNAzyme 2/sub-2, and the as-prepared artificial nanochannels were designed to be switchable between four different permeability states, including state 1 (hydrophobization, negative rectification), state 2 (hydrophilization, negative rectification), state 3 (hydrophobization, positive rectification), and state 4 (hydrophilization, positive rectification). The corresponding $I–V$ curves under the four different states were tested (Fig. 4a). The ICR ratio of state 1 was 3.36, indicating a negatively charged inner surface. After $Mg^{2+}$ was added, the ICR ratio changed from 3.36 to 1.90 due to the falling out of sub-1 with cholesterol tail, and the nanochannel was then switched from state 1 to state 2. The ICR ratios of state 3 and state 4 were further decreased to 0.87 and 0.53, respectively, which could be attributed to the exposure of amino groups upon the addition of $Zn^{2+}$. From Nyquist plots (Supplementary Fig. 16a), it could be observed that state 1 showed a higher $R_{ct}$ than state 2, while state 4 represented the minimum $R_{ct}$ value. The contact angle at different states (state 1, state 2, state 3 and state 4) was detected to be ($47.31 \pm 2.13°$), ($26.74 \pm 1.84°$), ($46.32 \pm 1.15°$) and ($36.75 \pm 1.57°$) respectively (Supplementary Fig. 16b), which confirmed the interconvertible hydrophobicity of the nanochannels, since higher contact angles indicated less wetting surfaces. The above results demonstrated that the nanochannel could be reversibly switched between these four different states. Next, the reversible conversion between any two different states was tested and observed (Fig. 4b–g). The above results demonstrated the capability of DNAzyme-based nanochannels to achieve accurate and reversible control of inner surface states, which was also verified by gel analysis (Supplementary Figs. 17 and 18).

## Controlled directional flow in artificial nanochannels from outside to cytoplasm

Here, we chose three different dyes, FITC, MB, and trypan blue (TB, an anionic dye used for revealing cell viability), to test the permeability of

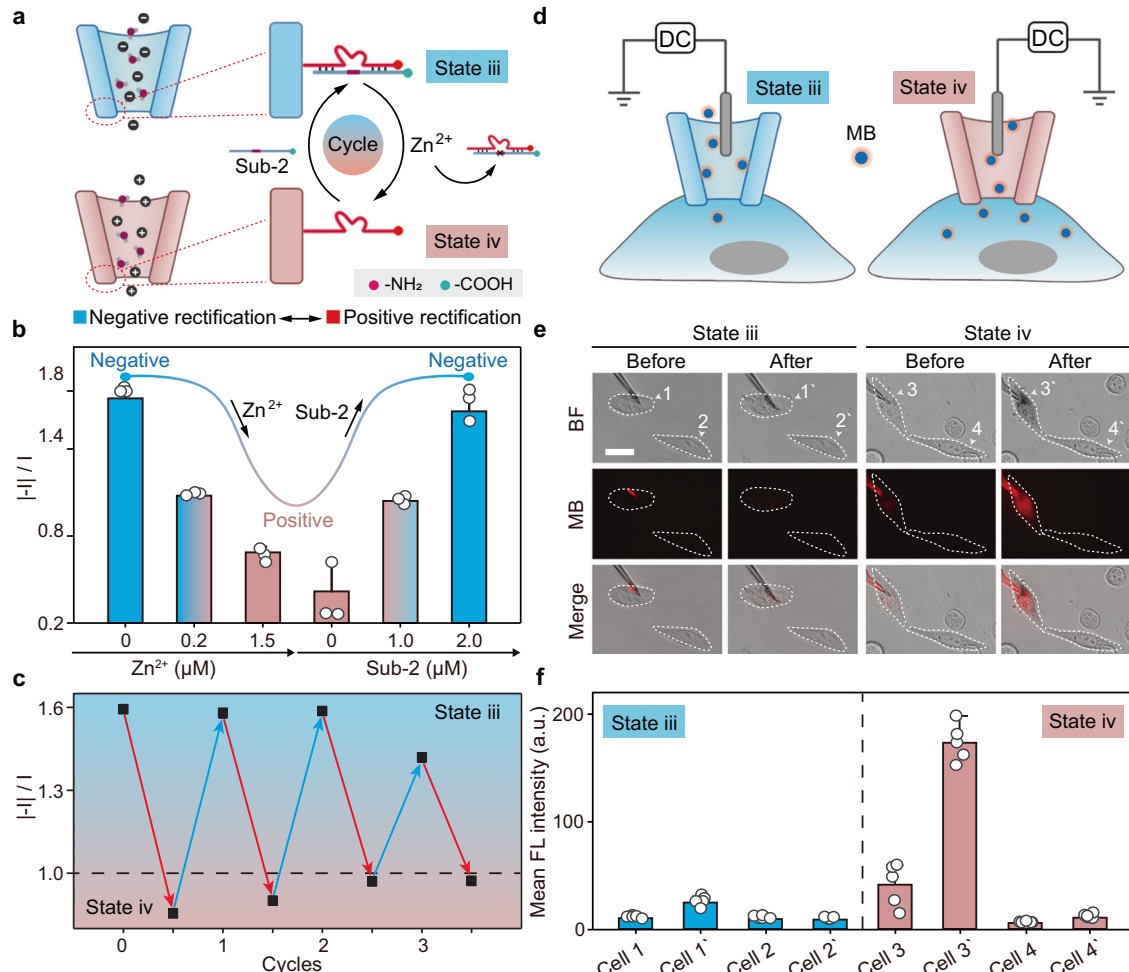

**Fig. 3 | Reversible charge control in a nanopipette. a** Design of a DNAzyme-based switch (DNAzyme 2 and substrate 2) for inner surface charge control in a nano-pipette (charge-reversal nanochannel). State iii: negative. State iv: positive. **b** Current rectification ratio measured in a charge-reversal nanochannel upon the treatment of different concentrations of $Zn^{2+}$ and substrate 2 (sub-2). **c** The reversibility of the charge-reversal nanochannel. Downward arrow: $Zn^{2+}$ treatment. Upward arrow: sub-2 treatment. **d** Schematic illustration showing transport of MB in a charge-reversal nanochannel under a DC voltage into single living HeLa cells.

**e** Microscopic cell images showing the outflow of MB into single living HeLa cells. BF bight-field. MB red fluorescence. Merge: mixed red & bright-field channel. Scale bar: 20 μm. **f** FL intensity comparison histogram of MB in **e**. The error bars came from 5 parallel measurements. Error bars in **b** represent the standard deviation of three independent experimental repeats, and the measure of the center represents their corresponding mean value. Error bars in **f** represent the standard deviation of five independent experimental repeats, and the measure of the center represents their corresponding mean value. Source data are provided as a Source Data file.

the functionalized nanochannels under different states. As shown in Fig. 5a, b, the FL of intracellular FITC after injection was significantly higher in cell 2' and cell 4', indicating an increase lipid permeability in state 2 and state 4. The FL of intracellular MB after injection was obviously higher in cell 3' and cell 4', indicating an increased cation permeability in state 3 and state 4 (Fig. 5c, d). The FL of intracellular TB after injection was obviously higher in cell 1' and cell 2', indicating an improved anion permeability in state 1 and state 2 (Fig. 5e, f). It could be clearly observed that the inner surface of the nanochannel in state 1 was less attractive to TB molecules, resulting in increased transport of TB into cells. The inner surface in state 2 was less attractive to FITC and TB molecules after the cleavage of cholesterol groups, leading to the significantly increased transport of FITC and TB into cells. In state 3, sub-2 strand was cleaved, causing immediate exposure of amino groups, which was less attractive to MB molecules. In state 4, the falling out of sub-1 decreased the molecular attraction of the inner surface to FITC and MB molecules, resulting in increased transport of FITC and MB into cells (Fig. 5g, h). The injection processes of three dyes through different nanochannels with different states were shown in Supplementary Movies 1–13 and Supplementary Figs. 19–21. In addition,

control experiments (bare nanochannel without the modification of DNAzymes) were also performed to study the transport of the three different dyes in Supplementary Figs. 22–24, which further demonstrated that the selective transport was mainly influenced by the wettability and charge of the inner wall of the nanochannel.

## Evaluation and simulation of nanopipette-based transmembrane delivery

To mimic physiological conditions, we embedded the artificial nanochannels into the plasma membrane of a single living HeLa cell using the truncated tip of the nanopipettes, which showed selective transport of TB dyes through the living cell membrane (TB dye molecules can stain dead cells, but cannot penetrate the membranes of living cells), just like real biological channels without external potential. Numerical simulations based on the commercial finiteelement software package COMSOL Multiphysics (version 5.4) were performed to investigate the delivery mechanism of TB dyes in an artificial nanochannel. The simulated geometry of the nanopipette is shown in Supplementary Figs. 25 and 26 (L, the length of the nanopipette; $d_1$, $d_2$, the diameter of the nanopipette tip and top, respectively; $d_3$, the thickness of the nanopipette wall). And the

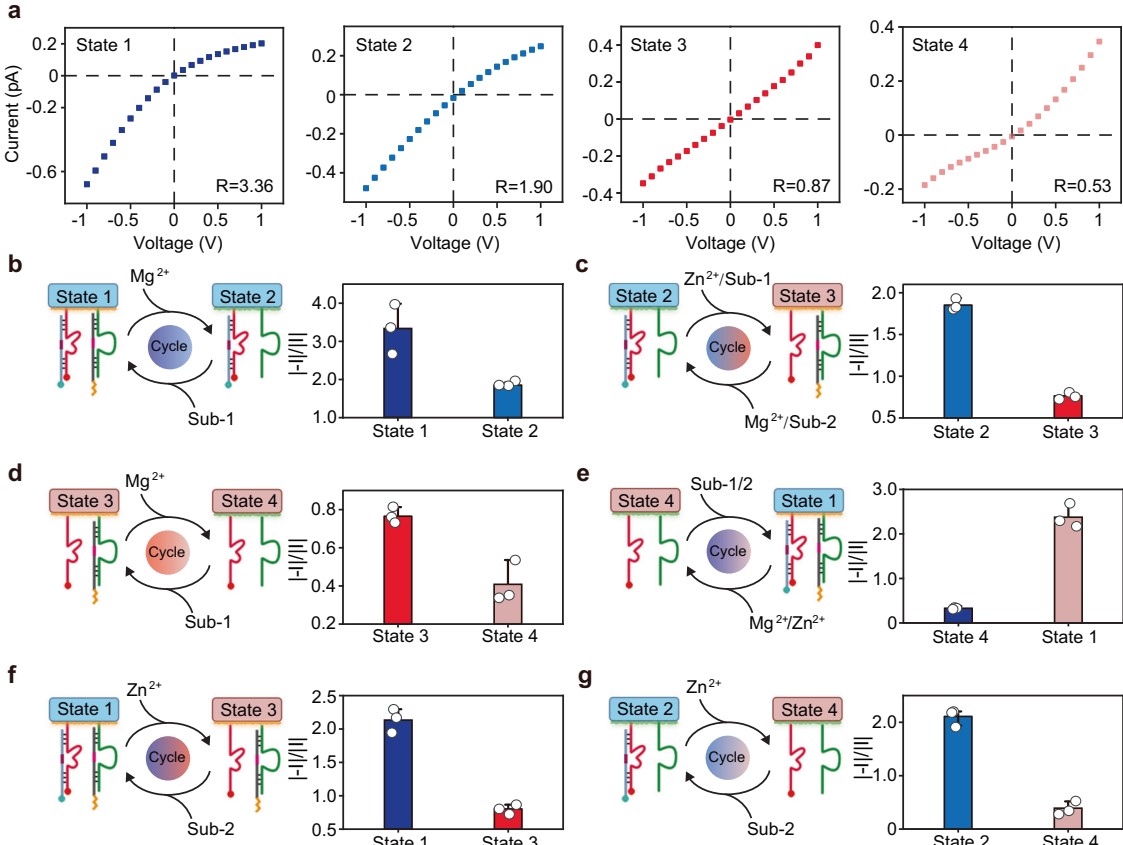

**Fig. 4 | Rectifying artificial nanochannels with four interconvertible permeability states. a** $I–V$ curves measured in an as-prepared nanochannel corresponding to state 1, state 2, state 3 and state 4. **b** Left: Schematic illustration showing the reversal switch between state 1 and state 2. Right: Current rectification ratio measured at state 1 and state 2. **c** Left: Schematic illustration showing the reversal switch between state 2 and state 3. Right: Current rectification ratio measured at state 2 and state 3. **d** Left: Schematic illustration showing the reversal switch between state 3 and state 4. Right: Current rectification ratio measured at state 3 and state 4. **e** Left: Schematic illustration showing the reversal switch between state 4 and state 1. Right: Current rectification ratio measured at state 4 and state 1. **f** Left: Schematic illustration showing the reversal switch between state 1 and state 3. Right: Current rectification ratio measured at state 1 and state 3. **g** Left: Schematic illustration showing the reversal switch between state 2 and state 4. Right: Current rectification ratio measured at state 2 and state 4. Error bars in **b–g** represent the standard deviation of three independent experimental repeats, and the measure of the center represents their corresponding mean value. Source data are provided as a Source Data file.

detailed simulation parameters of the nanopipette model are shown in Supplementary Table 4. The boundary conditions are shown in Supplementary Table 5. Despite the absence of external potential, the mutual interaction between the charge of TB dye molecules and DNAzyme (functionalized on the inner wall of nanopipette) still generated an electrostatic field. Supplementary Fig. 27 shows the distributions of potential and electric field inside the nanopipette. In states 1 and 2, the negatively charged inner surface endowed a repelling force to TB dye molecules, which was favorable for the delivery of TB. In contrast, in states 3 and 4, TB molecules were absorbed on the inner surface of the positively charged nanopipettes. Next, the difference in hydrophobic and hydrophilic performance between states 1 and 2 was taken into consideration. The cholesterol group modified on substrate 1 led to a hydrophobic inner surface in state 1, which was detrimental to the infiltration of hydrophilic TB. Thus, compared with state 1, state 2 with a hydrophilic inner surface was more favorable for presumably the transport of TB. The above factors determined the transmembrane delivery of TB, as it could be deduced that the TB transport rate was highest in state 2, followed by state 1, as reflected by the concentration gradient distribution in different states (Fig. 6a). From cell experiments (Fig. 6b), negligible FL of TB was observed in cells upon the presence of state 3 and 4. In contrast, strong FL was seen in states 1 and 2, and state 2 provided the strongest FL of TB (Supplementary Fig. 28). The above results were consistent with the previous simulation (Fig. 6a–c). As a

control, cells in different states were tested by TB staining. The FL of TB could not be observed in living cells with intact cell membranes (Supplementary Fig. 29), which indicated that the TB molecules were delivered through the embedded nanochannels in our previous experiments.

## Controlled gene silencing of miR-21 and Ca²⁺ influx based on rectifying artificial nanochannels

We further discussed the possible application of artificial nanochannels in the controlled regulation of cell functions or behaviors. First, the nanochannels were applied to study gene regulation in single-living HeLa cells. An antisense strand against microRNA-21 (miR-21) with a hairpin structure (hp-seed, labeled by FAM and its corresponding quencher, BHQ) was filled inside a nanopipette (state 1) for detecting and silencing miR-21. As shown in Fig. 7a, the FL of FAM was quenched by the adjacent BHQ group via the hairpin structure of the antisense strand. A DC voltage was applied to inject the antisense strand into a single HeLa cell, as the loop part of the strand was opened in the presence of miR-21 and led to the recovery of green FL of FAM, thus inducing the hybridization between the antisense strand and miR-21 (Supplementary Fig. 30). As shown in Fig. 7b, green FL could be obviously observed after injection, which indicated the antisense strand induced miR-21 recognizing and silencing (Supplementary Fig. 31). According to previous studies, miR-21 silencing can inhibit cancer proliferation, invasion, and metastasis, promote cell apoptosis,

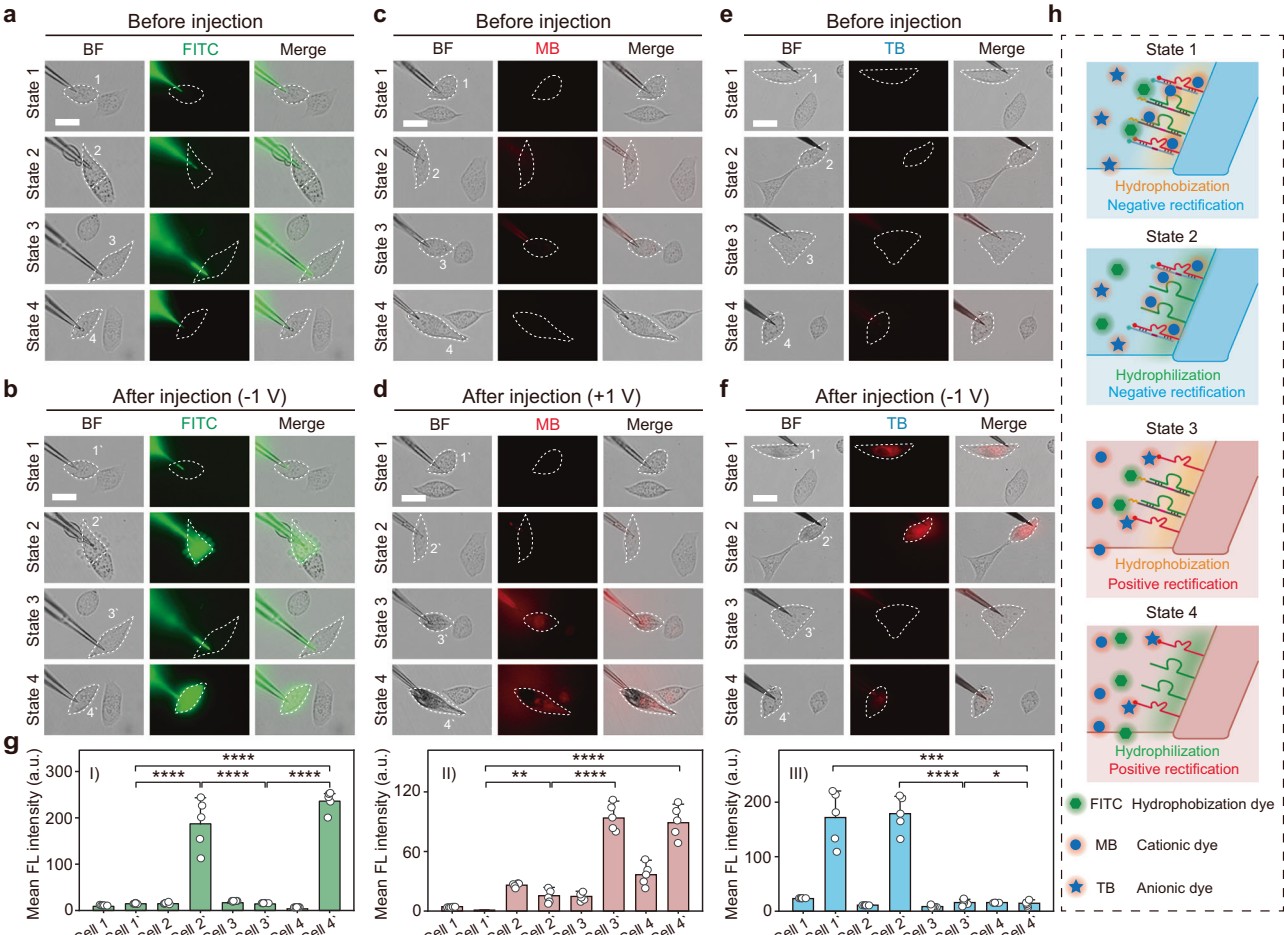

**Fig. 5 | Controlled directional flow in artificial nanochannels.** Microscopic cell images showing the outflow of FITC (**a**, **b**), MB (**c**, **d**), and TB (**e**, **f**) into single living HeLa cells before and after injection. BF bight-field, FITC green fluorescence, MB red fluorescence, TB red fluorescence, Merge mixed green/red & bright-field channel. **g** Left: FL intensity comparison histogram of FITC before and after injection. Medium: FL intensity comparison histogram of MB before and after injection. Right: FL intensity comparison histogram of TB before and after injection. Significance of the mean FL intensity was assessed by $t$ test ($n = 5$ independent experiments, and the data were presented as the mean values ± SDs, *$p < 0.05$; **$p < 0.01$; ***$p < 0.001$; ****$p < 0.0001$). All statistics were calculated using two-tailed paired $t$ test. **h**, Schematic illustration showing the interaction between different dyes and the inner surface of the nanochannel under the four different states. Scale bars: 20 μm. Error bars in **g** represent the standard deviation of five independent experimental repeats, and the measure of the center represents their corresponding mean value. Source data are provided as a Source Data file.

and reduce the resistance of tumor cells to chemotherapy[48]. As shown in Supplementary Fig. 32, HeLa cells treated with seed represented inhibited proliferation after 12 h. After 24 h, obvious apoptosis of HeLa cells was observed. Next, an artificial nanochannel (state 3) was embedded into the plasma membrane of a single living PC-12 cell, which showed selective transport of $Ca^{2+}$ through the living cell membrane (Fig. 7c). A FL probe, Fluo-4, was used for detecting $Ca^{2+}$ in cells[49]. As shown in Fig. 7d, after the treatment of $Ca^{2+}$ (1 mM), the FL of Fluo-4 was obviously increased, which indicated the influx of $Ca^{2+}$ through the nanochannel (Supplementary Fig. 33). Next, a nanochannel in state 1 was embedded into the plasma membrane of a single living PC-12 cell, after the addition of $Zn^{2+}$ (which can transfer the state of the nanochannel from state 1 to state 3), the FL of Fluo-4 was enhanced in 20 min. This result indicated that the DNAzyme-functionalized nanochannel can be artificially regulated to switch from "close" to "open" for the selective transport of $Ca^{2+}$ (Supplementary Fig. 34). In contrast, the FL intensity of Fluo-4 in the control group (a single PC-12 cell without any modification or interference) was gradually decreased, which could be attributed to the intracellular degradation of dye molecules quenched by laser light (Supplementary Fig. 35). Spontaneous FL of HeLa cells and PC-12 cells not significant

(Supplementary Fig. 36). Besides, $Ca^{2+}$ is an important signaling molecule within neuron cells and is involved in many biological processes such as neurotransmitter release and apoptosis. Changes in $Ca^{2+}$ concentration can induce excitatory or inhibitory responses in neurons, thus affecting the transmission of neural signals[50]. Therefore, we have demonstrated the exciting potential of the DNAzymes-functionalized artificial nanochannel for cellular function regulation at the single-cell level.

## Discussion

The design of highly controlled nanochannels has been a longstanding goal in biomimetics and cellular research. To tackle this question, we prepared DNAzyme-functionalized artificial biomimetic nanochannels based on glass nanopipettes to realize delicate control of channel permeability, which enabled precise regulation of the inner surface wettability and charge through two pairs of DNAzyme-substrate molecular controlled switches. The permeability of the rectifying artificial nanochannels could be reversibly switched between four different states: state 1 (hydrophobization, negative rectification), state 2 (hydrophilization, negative rectification), state 3 (hydrophobization, positive rectification), and state 4 (hydrophilization,

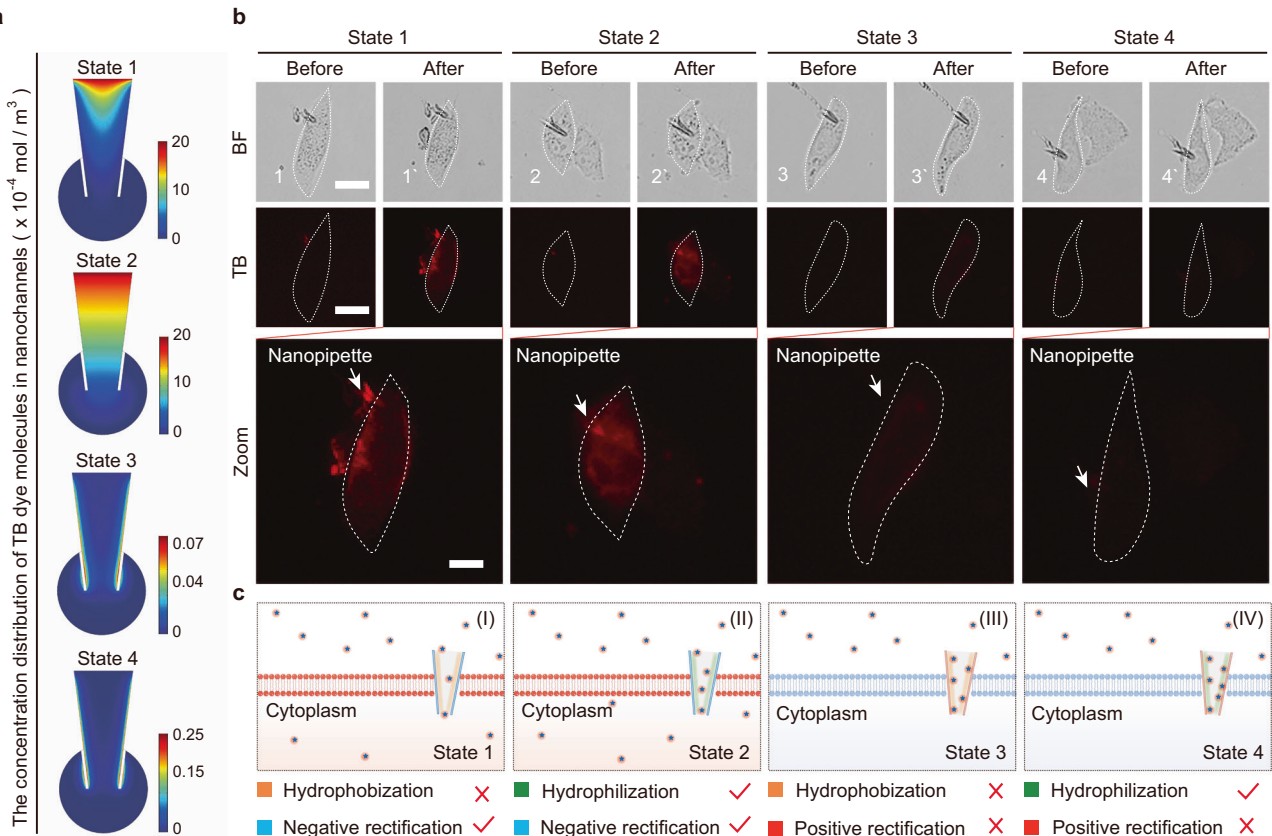

**Fig. 6 | Nanopipette-based transmembrane delivery. a** The concentration distribution of TB dye molecules between the top to the tip of the nanochannels. **b** Microscopic cell images showing the transport of TB dye molecules into single living HeLa cells before and after the treatment of TB dyes. BF bight-field, TB: red fluorescence. Merge: mixed red & bright-field channel. Experiments were repeated five times independently with similar results. **c** Schematic illustration showing the transport process of TB dye molecules through nanochannels embedded in living cell under different states, corresponding to b. State 1: hydrophobization, negative rectification. State 2: hydrophilization, negative rectification. State 3: hydrophobization, positive rectification. State 4: hydrophilization, positive rectification. A tick indicated a favorable condition for the delivery of TB. A 'X' indicated an unfavorable condition for the delivery of TB. Scale bar: 20 μm. Scale bar in zoom area: 10 μm.

positive rectification), which exhibited distinct transport patterns to different functional molecules and bioactive ions. Upon the embedding of the artificial nanochannels into living cell membranes, we further showed selective transport of TB dye molecules across the plasma membrane. Finally, we achieved selective delivery of antisense strands for gene regulation in single HeLa cells. Using PC-12 cells, we also realized a controlled influx of $Ca^{2+}$ ions. Since a wide variety of metal-specific DNAzymes are available, our method can be readily applied to construct other dynamic regulation switches controlled by other metal ions inside artificial nanochannels, providing a smart and versatile platform for the design of rectifying artificial nanochannels with on-demand functions.

In the future, efforts will be dedicated to several potential applications. For example, if the inner surface of rectifying artificial nanochannels is pH-sensitive with specific selectivity for metal ions, the pH-related activity of metal ion channels in different organelles can be detected and modulated. Another example is to use artificial nanochannels to tune the membrane potential of biologically significant cells, such as neural cells, as the selective influx of active ions may be useful for the manipulation of cell behaviors.

## Methods
### Ethical statement
Our research does not include any studies involving humans or animals.

## Materials
HeLa cervical cancer cells and rat pheochromocytoma 12 (PC-12) cells were purchased from bluefbio (Shanghai) Biology Technology Development Co., Ltd. (Shanghai, China). $ZnCl_2$, $MgCl_2$, KCl, 30% $H_2O_2$, 98% $H_2SO_4$, (3-aminopropyl)triethoxysilane (APTES), 25% Glutaric dialdehyde (GA) solution and Fluo-4 AM probe were purchased from Adamas-beta® (Shanghai, China). Fluo-4 AM probe is an FL probe for detecting the concentration of $Ca^{2+}$ in cells, which would be retained in cells with Fluo-4 state after entering the cells and cutting by esterase. $K_3[Fe(CN)_6]$ and $K_4[Fe(CN)_6]$ were purchased from Macklin (Shanghai, China). All other chemical reagents were of analytical grade and purchased from Sigma-Aldrich, Inc. (USA). PBS buffer (pH 7.4) contained 136.7 mM NaCl, 2.7 mM KCl, 8.72 mM $Na_2HPO_4$, and 1.41 mM $KH_2PO_4$. All solutions were prepared using ultrapure water obtained through a Millipore Milli-Q water purification system (Billerica, MA, USA), with an electric resistance of >18.2 MΩ. FITC dyes, methylene blue dyes (MB), Trypan blue dyes (TB) were purchased from Energy Chemical (Shanghai, China). FITC is lipophilic, which can form hydrophobic interaction with cholesterol groups at the end of sub-1. To mimic controlled directional flow in natural transmembrane channels, FITC was selected to test the permeability of the functionalized nanochannels switchable between hydrophobic and hydrophilic states. In addition, MB, a commonly used cationic dye, as well TB, a typical anionic dye, were selected to test the permeability of the functionalized nanochannels switchable between negative and positive rectification. All DNA sequences were purchased from Sangon Biological

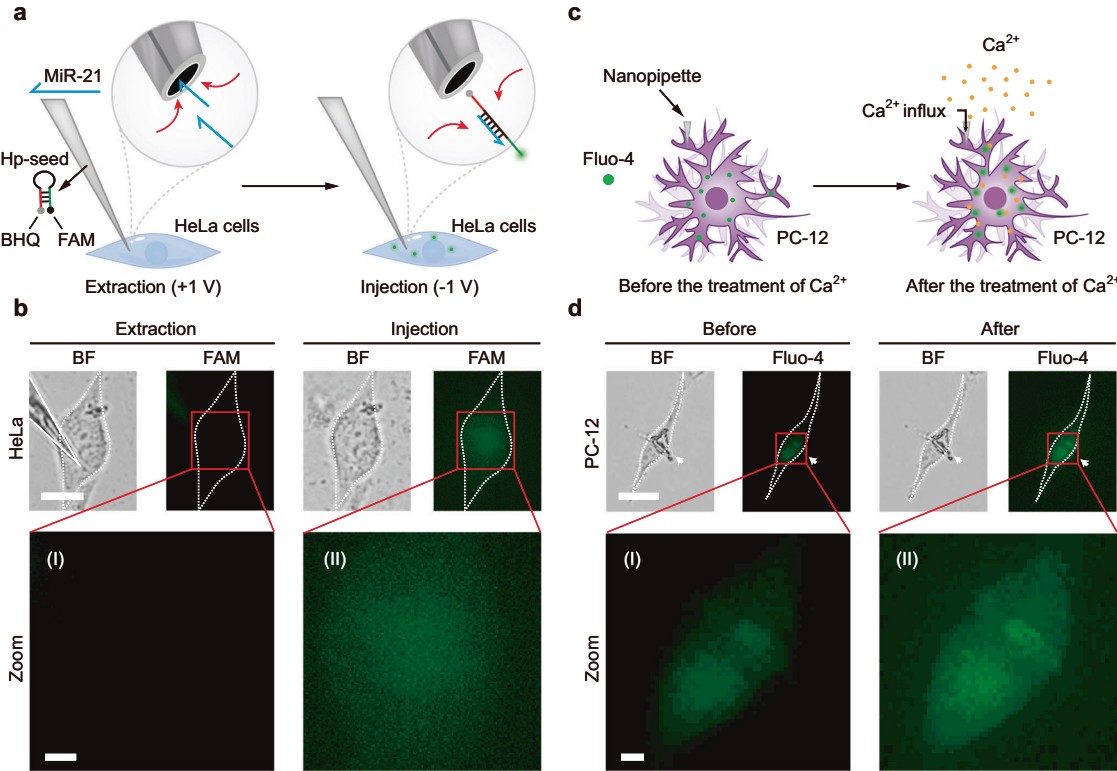

**Fig. 7 | Controlled gene regulation and Ca²⁺ influx. a** Schematic illustration showing the artificial nanochannel (state 1) for miR-21 silencing in single living cell. **b** The nanopipette was embedded into a living HeLa cell. Upon application of a DC voltage (+1 V, 1 min), the miR-21 was extracted into to tip. After hybridization between miR-21 and FAM-labeled antisense hairpin strand (hp-seed), the silenced miR-21 labeled by FAM was delivered into the cell via injection (−1 V, 2 min). BF bight-field. FAM: green fluorescence. Merge: mixed green & bright-field channel.

Experiments were repeated three times independently with similar results. **c** Schematic illustration showing Ca²⁺ influx based on the artificial nanochannel (state 3). **d** A nanopipette tip was embedded in a living PC-12 cell. The cell was stained by Fluo-4. The fluorescence of the cell was detected before and after the treatment of Ca²⁺. Experiments were repeated three times independently with similar results. BF: bight-field. Fluo-4: green fluorescence. Merge: mixed green & bright-field channel. Scale bars: 20 μm, scale bars in zoom area: 5 μm.

Engineering Technology Co., Ltd. (Shanghai, China). The sequences were listed in Supporting Information (Supplementary Table 2).

## Instrumental methods

All the FL images were obtained using a water dipping objective (×60) on a laser scanning confocal microscope (Nikon A1R., Japan). All reagents were weighed by an analytical balance (ME 104, Mettler Toledo). All reagents were centrifuged by a centrifuge (Centrifuge 5430, Eppendorf). Scanning electron microscopy (SEM) images of nanopipette morphology were performed using a field emission scanning electron microscope (Ultra 55, Carl Zeiss Ltd., Germany). The nanopipette was fixed under the microscope for electrochemical measurements by a holder (Axon Instruments, Union City, CA) connected to an Axopatch 200B low-noise amplifier and an Axon Digidata 1550 A low-noise data acquisition system (Molecular Devices, Sunnyvale, CA). The FL spectra were gained on an FL spectrophotometer (LS-55 Lumine). UV-vis spectrophotometer (Ocean Optics Inc., USA) was used to measure the ultraviolet absorbance. Electrochemical impedance spectroscopy (EIS) was done on a CHI660E electrochemical workstation (CHI660E, Shanghai Chenhua Co., LTD., China). A conventional three-electrode system consisting of the working electrode (bare or modified planar glass plates), the reference electrode (Ag/AgCl) and the counter electrode (platinum wire) was employed.

## Cell culture

HeLa cells were cultured in DMEM medium supplemented with 10 % fetal bovine serum, and penicillin/streptomycin (100 units/mL). PC-12 cells were cultured in RPMI-1640 media with 10% horse serum, 5% fetal bovine serum and 2 mM Glutamine. The culture dishes were placed in a humid atmosphere at 37 °C with 5 % CO₂.

## Preparation of laser-pulled nanopipettes

The borosilicate glass capillaries (BF100-58-10, Sutter Instrument Co., Novato, CA, USA), with an outer diameter of 1.0 mm and inner diameter of 0.50 mm, were thoroughly cleaned with piranha solution for 30 min to remove organic impurities on the surface before pulling. The nanopipettes used in this work were fabricated using a Sutter P-2000 laser puller, and the parameters of the nanopipettes were regulated by a one-step program, as shown in Supplementary Table 3. The variation of nanopipette pulling times was controlled within 0.1 s to ensure the reproducibility of the aperture geometry.

## DNAzyme modification on the inner surface of nanopipettes

We used two different ways to achieve DNAzyme modification. The first method was to induce aldehyde groups on the glass surface to bind with amino group-labeled DNAzyme (NH₂-DNAzyme, Supplementary Fig. 1). First, 5% aminopropyl triethoxysilane (APTES) solution (prepared by ethanol) was injected into the tip of nanopipettes, then washed with ethanol and dried in a vacuum drying oven at 110 °C to obtain a continuous silanized film. Then 2.5% glutaric dialdehyde (GA) solution (prepared by ultrapure water) was injected into the tip of nanopipettes for 10 h. After washing, the DNAzyme solution was injected into the nanopipettes for 6 h and then filled with PBS buffer solution. The second method was to induce Au cladding on the glass surface to bind with thiol-labeled DNAzyme (SH-DNAzyme, Supplementary Fig. 2). Specifically, ethanol and chlorauric acid (HAuCl₄, 8 mM) were mixed according to the volume ratio of 1:3 and then

injected into the tip of nanopipettes for 2.5 h under 254 nm UV light irradiation. After drying in a vacuum drying oven at 100 °C, a continuous gold film was obtained. Then the mixed solution of SH-labeled DNAzyme and tris(2-carboxyethyl)-phosphine (TCEP) was filled into tip of the nanopipettes for 12 h to achieve DNAzyme ligation.

### Reversible wettability/charge control of the DNAzyme-functionalized glass nanopipettes between four different states

After the modification of two pairs of double-chain DNAzymes ($Mg^{2+}$/$Zn^{2+}$-specific) on the inner wall of the nanopipettes (state 1), these functionalized nanopipettes were filled with electrolyte solution and inserted by an Ag/AgCl electrode for the characterization of rectification inside the artificial nanochannels. Then the nanopipettes were filled with $Mg^{2+}$ (2 µM) for 20 min to achieve the cleavage of sub-1 in order to release cholesterol tail (state 2). Then the nanopipettes were refilled with electrolyte solution and sent for characterization after 2 times of washing. The addition of sub-1 solution (1 µM) could transfer the state of the nanochannels from state 2 to state 1. Similarly, the reversible transfer from state 2 to state 4 was achieved by adding $Zn^{2+}$ (2 µM). For the switch from state 4 to state 2, sub-2 (1 µM) was added. $Mg^{2+}$ and sub-1 were used to perform reversible transfer between state 3 and state 4. $Zn^{2+}$ and sub-2 were used to perform reversible transfer between state 1 and state 3. $Mg^{2+}$/$Zn^{2+}$ and sub-1/sub-2 were used to perform reversible transfer between state 1 and state 4. $Mg^{2+}$/sub-2 and $Zn^{2+}$/sub-1 were used to perform reversible transfer between state 2 and state 3.

### Controlled $Ca^{2+}$ influx into PC-12 cells

First, PC-12 cells were washed with PBS buffer (pH 7.4) three times, then treated with Fluo-4 AM probe (1 µM, dissolved in PBS buffer) for 30 min at room temperature. After washing with PBS for 3 times, cells were cultured for another 20 min at 37 °C, allowing the transformation of Fluo-4 AM to Fluo-4. Next, an artificial nanochannel (state 3) was embedded into the plasma membrane of a single living PC-12 cell using the truncated tip of the nanopipettes. Then the culture medium was replaced by PBS containing 1 mM of $Ca^{2+}$, and the cell was sent for microscopic observation after 20 min. In order to test the artificial controlled "close" and "open" of the nanochannel, the tip of the nanopipette in state 1 was embedded into the cell membrane of a single living PC-12 cell, and the culture medium was replaced by PBS containing 1 mM of $Ca^{2+}$. 1 mM of $Zn^{2+}$ was then added into the culture medium and incubated for 20 min to achieve the transfer of the nanopipette state from state 1 to state 3. The whole process was observed under a microscope.

### Gel electrophoresis analysis

The gel was prepared in 1 × TBE buffer and was run at 80 V for 1 h. The agarose gel was first run in 1 × Tris-Borate-EDTA buffer at 100 V for 35 min and then stained by ethidium bromide. Finally, the gel was imaged using a Bio-Rad molecular imager under blue light.

### ICR analysis

Two Ag/AgCl wires were introduced for obtaining electrochemical signals, as one of them was used as the working electrode inserted into the nanochannel filled with electrolyte solution (0.1 × PBS, pH 7.4), and the other one was used as the reference electrode bathed in DMEM solution. Before electrochemical measurements, the liquid-filled tip of the nanochannel was centrifuged for the exclusion of the air bubbles. Then the nanopipette was fixed on a holder and connected to the headstage of the Axopatch 200B device. A previously designed protocol was edited to perform electrochemical experiments with pClamp 10.7 (Axon Instrument, Forest City, USA) run on a PC. The record mode was gap-free with a sampling frequency of 100 kHz and a 5 kHz low-pass Bessel filter.

### EIS measurement

EIS analysis was conducted in 10 mM $[Fe(CN)_6]^{3-}$/$[Fe(CN)_6]^{4-}$ dissolved in 1 × PBS (pH 7.4) containing 0.1 M KCl at room temperature. The experimental parameters were as follows: open circuit potential, +0.25 V; potential amplitude, ±10 mV; frequency range, 0.1–100,000 Hz.

### Contact angle measurement

Contact angle (CA) measurements were performed via an OCA 20 system to collect the CA data of planar glass plates at different states: 1. without any modification (control); 2. with the modification of DNAzyme 1/Sub-1(without and with the treatment of $Mg^{2+}$, state (i/ii); 3. with the modification of DNAzyme 2/Sub-2(without and with the treatment of $Zn^{2+}$, state (iii/iv), 4. with the modification of mixture solution of DNAzyme 1/Sub-1 and DNAzyme 2/Sub-2 (state 1); 5. with the modification of mixture solution of DNAzyme 1 and DNAzyme 2/Sub-2 (state 2); 6. with the modification of mixture solution of DNAzyme 1/Sub-1 and DNAzyme 2 (state 3); 7. with the modification of mixture solution of DNAzyme 1 and DNAzyme 2 (state 4). The average CA value was obtained from three different locations.

### Finite element simulations

Numerical simulations of the electro-osmotic flow inside the nanopipette were performed using COMSOL Multiphysics 5.4 (COMSOL AB, Stockholm, Sweden) to show the detailed profiles of the voltage and concentration distribution of TB dyes. The 2D axisymmetric model was designed to mimic the cone-shaped nanopipettes based on the electrostatics module and the creeping flow module. For the simulation of FL molecules delivered into cytoplasm, the finite element method model was built using the parameters as shown in Supplementary Figs. 25 and 26, Supplementary Tables 4 and 5.

### Statistics and reproducibility

The sample analysis, at least 3 samples were tested for each state (state 1, state 2, state 3, and state 4). No statistical methods were used to determine sample size. The sample size was decided according to a previous study using the same cohorts of samples. The selection of the sample size took into account both statistical significance and the timeliness of validating experimental performance in exploratory experiments. No data were excluded from the analysis. Experiments were repeated three independently times. All attempts of replication were successful. Cells were randomly allocated into control and experimental groups for cell injection. Data collection and analysis were not blinded. Blinding was not relevant because the study is a pilot investigation. The main purpose of the study was to test the feasibility of DNAzyme-functionalized artificial biomimetic nanochannels to realize precise control of the inner surface wettability and charge between four different states under external stimuli. Blinding might complicate the experimental setup or workflow, potentially leading to inefficiencies in this pilot experiment.

### Reporting summary

Further information on research design is available in the Nature Portfolio Reporting Summary linked to this article.

## Data availability

The main data supporting the results of this study are available within the paper and its Supplementary Information. The raw data have been provided in source data and have been deposited in Figshare database at https://figshare.com/s/7f0e66ea96f7ac4f557a. Additional relevant information is available from the corresponding author. Source data are provided in this paper.

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

## Acknowledgements

This research was supported by the National Natural Science Foundation of China (21977031), Science and Technology Commission of Shanghai Municipality (2018SHZDZX03), Shanghai Science and Technology Committee (22ZR1416800, 23ZR1416100), and the Fundamental Research Funds for the Central Universities to R.Q., and the U.S. National Institutes of Health (GM141931) to Y.L. Thanks eceshi (www.eceshi.com) for the Zeta analysis.

## Author contributions

R.C.Q. and Y.L. conceived the project. M.S.W., Z.R.Z., X.Y.W., D.W.L. and R.C.Q. performed the experiments and analyzed the results. R.C.Q., M.S.W., W.G., Y.W., Z.Y. and Y.L. wrote the manuscript.

## Competing interests

The authors declare no competing interests.
