## [Peer Review File · Nature Communications]

Rectifying artificial nanochannels with multiple
interconvertible permeability statesReviewers' Comments:

Reviewer #1 (Remarks to the Author)

This work entitled “Rectifying artificial nanochannels with multiple interconvertible permeability states” described a convenient method to build artificial biomimetic nanochannels based on DNAzyme-functionalized glass nanopipettes, which realized delicate control of channel permeability. Metal ion-specific RNA-cleaving DNAzymes and their substrates were used for modification of glass nanopipettes, enabling the manipulation of wettability and charge at the inner surface under external stimuli. The authors used the artificial nanochannels to perform selective transport across cell membrane, allowing cellular function regulation at single-cell level. The manuscript is clearly presented and the findings look convincing and well supported by the data. Few minor revisions need to be addressed before its publication.

1. Ca^{2+} ions play important roles in tuning cell functions. Related descriptions of Ca^{2+} should be added in the introduction part.
2. As inspired from the biological channels, the authors mentioned only in the introduction part. I would recommend to focus more on the recent works about the bio-inspired artificial nanochannels.
3. Future work should be lined and specified in detail to show this work application to a broad audience.
4. It is better to introduce the chemical properties of fluorescent molecules used for demonstrating their selective transport across the nanochannel.

Reviewer #2 (Remarks to the Author)

This paper presents the development of artificial nanochannels that exhibit four distinct permeable states, regulated by DNAzyme-functionality. This represents an advance on previous studies, which have been limited to controlling just two permeable states. The approach taken here contributes towards the mimicking of biological ion channels, and the realization of multiple triggered stimuli. The scheme is clearly described and supported by persuasive evidence. Various experimental results, including a demonstration of cell signaling through the proposed nanochannel, illustrate the different permeable states. However, this paper includes some critical controversial points that should be explained and cleared by the authors.

1. The most controversial point, in my opinion, is the proof of hydrophobicity of nanochannel and the related I-V characteristics and rectified signal. I also agree that the cholesterol tail would provide the hydrophobicity on nanochannel. However, I cannot understand the claims by the authors, i.e., as shown in Fig. 2b, ICR ratio decrease according to increase of Mg^{2+} is the proof of hydrophobicity. In general, the surface charge of hydrophobic surface is much lower than that of hydrophilic surface. It induces thinner EDL, which means the rectification provoked by the overlap of EDL becomes weaker. Therefore, the opposite ICR ratio change should happen if the authors' claim is correct.

2. Additionally, the result that, as shown in Fig.4, the current level of state 1 is higher than that of state 2, conflicts the hydrophobic barrier mechanism in page 5 claimed by the authors, I think that since the presented nanochannel size is much larger than the critical size, which is the diameter of an ion's first hydration shell, the hydrophobic barrier mechanism is not applicable. Rather, hydrophobic nanochannel can show high conductivity and flow compared to the hydrophilic nanochannel owing to the hydrodynamic slip[1-2].

[1] Kun Li et al., Anomalous ion transport through hydrophilic and hydrophobic nanopores, 2017 IEEE International Conference on Manipulation, Manufacturing and Measurement on the Nanoscale (3MNANO), 151-155 (2017).

[2] Olga I. Vinogradova et al, Transport of ions in hydrophobic nanotubes, Physics of Fluids 34, 122003 (2022).

3. Regarding this issue, the authors should suggest another proof way for hydrophobicity of nanochannel beside of change of rectification or clear explanation for these unexpected results.

4. Rectification of tapered nanochannel comes from the EDL overlap. So, highly concentrated medium can induce disadvantage for rectification. The medium in this study is PBS having high concentration of NaCl. The authors should check the theoretical Debye layer thickness is reasonable considering the geometrical size of nanochannel for the proper operation in this medium.

5. As far as I know, MB has no red fluorescent excitation (Maybe I wrong). Is there any fluorescent tagging for MB in this study? If not, can the author add the explanation about the origin of red fluorescent signal in MB?

Reviewer #3 (Remarks to the Author)

Drawing inspiration from transmembrane ion channels, the authors have engineered artificial nanochannels that mimic the selective permeation function of these biological channels. These artificial constructs emerge as promising tools with a multitude of potential applications in the realm of biomedical research.

In this study, the authors delineate a strategy for fabricating biomimetic nanochannels using glass nanopipettes, internally modified with DNAzymes capable of cleaving RNA. DNAzymes are specific DNA sequences exhibiting enzymatic activity that necessitate a particular metal ion cofactor for their function. Through this innovation, the authors successfully achieved reversible switching of the nanopipette's permeability across four distinct states. Moreover, upon integration of these nanochannels into living cells, they demonstrated selective transport of bioactive molecules.

To institute the four permeability states within the nanopipettes, their internal wall was modified using two specific DNAzymes: one targeting Mg^{2+} and the other Zn^{2+} . With these DNAzymes anchored, it became possible to modulate the nanochannel's permeability across four configurations: hydrophobic with negative rectification, hydrophilic with negative rectification, hydrophobic with positive

rectification, and hydrophilic with positive rectification.

Two methodologies were postulated for the internal modification of the pipettes: the induction of aldehyde groups and a gold coating. After evaluating the stability of both approaches, it was ascertained that the bond between the -CHO and -NH₂ groups was more robust, leading to the selection of this methodology for subsequent investigations.

Once these implementations have been achieved, the authors carry out a series of exciting demonstrations that allow evaluating Reversible wettability control in a nanopipette, Reversible charge control in a nanopipette, Rectifying artificial nanochannels with four interconvertible permeability states. The Evaluation and simulation of nanopipette-based transmembrane delivery. All these demonstrations are done using different electrophysiological tools that are well-documented and validated in the article. Although the article is very well implemented experimentally, I note that the main weakness of the article is conceptual. In the section titled "Design of artificial nanochannels based on DNzyme-functionalized nanopipettes," the description of biological channels is too generalized, and the channel concept is forced to emphasize the similarity of these biological entities with functionalized nanopipettes. Specifically, the pores of the channels are not all hydrophobic. Furthermore, the description of the selectivity filter differs from the reality of many known ion channels. For example, the K channels have particular selectivity filters, constituted by the backbone of a sequence of uncharged residues, TVGYG. Therefore, I request that the authors provide a more detailed and precise description of the biological entity that they used as a reference, which could be a specific type of ion channel or another transmembrane protein that better reflects the findings of the characteristics presented in this study.

Response Reviewers' Comments

We sincerely thank the reviewers for their valuable comments and constructive questions. According to the comments and editor's suggestion, we have revised the manuscript carefully. Here are our responses to reviewers' comments and changes we have made. All the changes are highlighted by yellow background in the revised manuscript.

Response to reviewer 1:

General Comment:

This work entitled "Rectifying artificial nanochannels with multiple interconvertible permeability states" described a convenient method to build artificial biomimetic nanochannels based on DNAzyme-functionalized glass nanopipettes, which realized delicate control of channel permeability. Metal ion-specific RNA-cleaving DNAzymes and their substrates were used for modification of glass nanopipettes, enabling the manipulation of wettability and charge at the inner surface under external stimuli. The authors used the artificial nanochannels to perform selective transport across cell membrane, allowing cellular function regulation at single-cell level. The manuscript is clearly presented and the findings look convincing and well supported by the data. Few minor revisions need to be addressed before its publication.

Reply:

We appreciate the reviewer's positive comments and valuable suggestions. In response to the comments, we have provided a detailed information, and revised the manuscript carefully to further support the conclusions and strengthen the manuscript. All the changes are highlighted by yellow background in the revised manuscript. The response and revision details are listed as follows:

Detailed Comments:

Comment 1: *Ca²⁺ ions play important roles in tuning cell functions. Related descriptions of Ca²⁺ should be added in the introduction part.*

Reply:

We appreciate the reviewer's comments and have followed the advice by rewriting the introduction part, as the related descriptions of Ca²⁺ has been added in Page 3, Lines 46-53 as following:

"Ca²⁺ ions are known to contribute to many biological processes and physiological activities of living cells, and is of great significance in maintaining the homeostasis of the body.¹² As a ubiquitous second messenger, Ca²⁺ ions can control various cellular biological functions, including energy metabolism, cellular differentiation, proliferation, survival and apoptosis.¹³ Therefore, Ca²⁺ channels play a vital role in the cell by allowing for the selective import and export of Ca²⁺ ions to modulate various cellular functions. This process is known as Ca²⁺ signaling, which is reported to be involved in various diseases, including cancer, autoimmune diseases and virus infection.¹⁴"

References:

12. Kong, F. Y., You, H. J., Zheng, K. Y., Tang, R. X. & Zheng, C. F. The crosstalk between pattern-recognition receptor signaling and calcium signaling. *Int. J. Biol. Macromol.* **192**, 745-756 (2021).
13. Monteith G. R., Prevarskaya N. & Roberts-Thomson S. J. The calcium-cancer signalling nexus. *Nat. Rev. Cancer* **17**, 373-380 (2017).
14. Clapham D. E. Calcium signaling. *Cell.* **131**, 1047-1058 (2007).

Comment 2: *As inspired from the biological channels, the authors mentioned only in the introduction part. I would recommend to focus more on the recent works about the bio-inspired artificial nanochannels.*

Reply:

We thank the reviewer for the suggestion and have added more recent references of bio-inspired artificial nanochannels in Page 3, Lines 53-57 as following:

“To construct biomimetic ion channels, various functional materials, such as hierarchically organized polymers, DNA nanopores, G-quadruplexes, metal-organic cuboctahedra and solid-state nanopores have been used.¹⁵⁻²⁰ These bio-inspired artificial nanochannels are designed to mimic selective transmembrane transport.”

References:

15. Reitemeier, J., Baek, S. & Bohn, P. W. Hydrophobic gating and spatial confinement in hierarchically organized block copolymer-nanopore electrode arrays for electrochemical biosensing of 4-ethyl phenol. *ACS Appl. Mater. Interfaces* **15**, 39707-39715 (2023).
16. Luo, L. et al. DNA nanopores as artificial membrane channels for bioprotonics. *Nat Commun* **14**, 5364 (2023).
17. Ikarashi, S. et al. DNA nanopore-tethered gold needle electrodes for channel current recording. *ACS Nano* **17**, 10598-10607 (2023).
18. Li, C. Y. Lipophilic G-quadruplex isomers as biomimetic ion channels for conformation-dependent selective transmembrane transport. *Anal. Chem.* **92**, 10169-10176 (2020).
19. Kawano, R. et al. Metal-organic cuboctahedra for synthetic ion channels with multiple conductance states. *Chem.* **2**, 393-403 (2017).
20. Yi, W. et al. Solid-state nanopore/nanochannel sensing of single entities. *Topics Curr Chem* **381**, 13 (2023).

Comment 3: *Future work should be lined and specified in detail to show this work application to a broad audience.*

Reply:

We appreciate the reviewer’s comments and have followed the advice by rewriting the conclusion part, as the future outlook including several potential applications has been added in the discussion section (Page 19, Lines 413-419) as following:

“In the future, efforts will be dedicated to several potential applications. For example, if the inner surface of rectifying artificial nanochannels is pH-sensitive with specific

selectivity for metal ions, the pH-related activity of metal ion channels in different organelles can be detected and modulated. Another example is to use the artificial nanochannels to tune the membrane potential of biologically significant cells such as neural cells, as the selective influx of active ions may be useful for the manipulation of cell behaviors.”

Comment 4: *It is better to introduce the chemical properties of fluorescent molecules used for demonstrating their selective transport across the nanochannel.*

Reply:

We agree with the reviewer and have provide more details of the fluorescent molecules in the material part (Page 20, Lines 433-439) as following:

“FITC is lipophilic, which can form hydrophobic interaction with cholesterol groups at the end of sub-1. To mimic controlled directional flow in natural transmembrane channels, FITC was selected to test the permeability of the functionalized nanochannels switchable between hydrophobic and hydrophilic states. In addition, MB, a commonly used cationic dye, as well as TB, a typical anionic dye, were selected to test the permeability of the functionalized nanochannels switchable between negative and positive rectification.”

Response to reviewer 2:

General Comment:

This paper presents the development of artificial nanochannels that exhibit four distinct permeable states, regulated by DNAzyme-functionality. This represents an advance on previous studies, which have been limited to controlling just two permeable states. The approach taken here contributes towards the mimicking of biological ion channels, and the realization of multiple triggered stimuli. The scheme is clearly described and supported by persuasive evidence. Various experimental results, including a demonstration of cell signaling through the proposed nanochannel, illustrate the different permeable states. However, this paper includes some critical controversial points that should be explained and cleared by the authors.

Reply:

We appreciate the reviewer’s comments, which contribute a lot to improve our manuscript. In response to the suggestions and questions, we have performed additional experiments, provided the corresponding discussions, and revised the manuscript carefully. All the changes are highlighted by yellow background in the revised manuscript.

Detailed Comments:

Comment 1. *The most controversial point, in my opinion, is the proof of hydrophobicity of nanochannel and the related I-V characteristics and rectified signal. I also agree that the cholesterol tail would provide the hydrophobicity on nanochannel. However, I cannot understand the claims by the authors, i.e., as shown in Fig. 2b, ICR ratio decrease according to increase of Mg^{2+} is the proof of hydrophobicity. In general, the*

surface charge of hydrophobic surface is much lower than that of hydrophilic surface. It induces thinner EDL, which means the rectification provoked by the overlap of EDL becomes weaker. Therefore, the opposite ICR ratio change should happen if the authors' claim is correct.

Reply:

We are sorry for not being clear in describing the ICR ratio results in Fig. 2b upon the addition of Mg^{2+} in the original submission. We agree with the reviewer that lower surface charge induces thinner EDL with weaker rectification. We also agree with the reviewer that in general, the surface charge of hydrophobic surface is lower than that of hydrophilic surface. Actually, in our work (Fig. 2a), the cholesterol tail was tagged at the end of sub-1. Upon hybridization between DNAzyme 1 (a Mg^{2+} -specific DNAzyme) and its substrate (sub-1), the inner surface of the nanochannel was modified by the hybridized DNA (dsDNA) initially (state i). When Mg^{2+} was added, the sub-1 strand was cleaved, and the nanochannel was then switched from state i to state ii, with its inner surface modification changed to single-stranded DNA (ssDNA). Due to the negative charge of DNA backbone, the dsDNA induced a more negatively charged surface with higher ICR ratio in state i than the ssDNA cleaved products in state ii (Supplementary Fig. 8a), because the conformation of the dsDNA exposed the negatively charged backbone to the surface more fully than the ssDNA.

In order to provide further experimental evidence on the difference of the inner surface charge between state i and state ii, we used electrochemical impedance spectroscopy for characterization. Nyquist plots for DNA modified surfaces before and after the addition of Mg^{2+} (2 μM) using 10 mM $[Fe(CN)_6]^{3-}/[Fe(CN)_6]^{4-}$ as redox mediators were obtained with a CHI660E electrochemical workstation. As shown in Supplementary Fig. 8b, the dsDNA modified surface (state i) showed a higher charge-transfer resistance (R_{ct}) than that of ssDNA modified surface (state ii), which indicated a more negative surface charge at state i. Therefore, although in general cases the surface charge of hydrophobic surface is lower, here the dsDNA at state i could provide more surface charges despite of the cholesterol tail.

The related descriptions have been added in Page 9, Lines 184-201 as following:

“In the initial stages of constructing the DNAzyme-based double control switches (Fig. 2a), we prepared nanochannels functionalized with a single pair of double chains composed of DNAzyme 1 (a Mg^{2+} -specific DNAzyme) and its substrate 1 (sub-1, tagged with cholesterol tail) to mimic the natural nanochannels which are switchable between dewetted (hydrophobic) and wetted states (hydrophilic). Upon hybridization between DNAzyme 1 and sub-1, the inner surface of the nanochannel was modified by the hybridized DNA (dsDNA) initially (state i). When Mg^{2+} was added, the sub-1 strand was cleaved, and the nanochannel was then switched from state i to state ii, with its inner surface modification changed to single-stranded DNA (ssDNA). Due to the negative charge of DNA backbone, the resulting dsDNA induced a more negatively charged surface with higher ICR ratio in state i than the ssDNA cleaved products in state ii (Supplementary Fig. 8a), because the conformation of the dsDNA exposed the negatively charged backbone to the surface more fully than the ssDNA. As shown in Fig. 2b, the addition of Mg^{2+} led to the cleavage of cholesterol tagged sub-1, thus the

inner surface of the nanochannel changed from state i containing dsDNA to state ii containing ssDNA. The dsDNA at state i induced a more negatively charged surface, which showed a higher charge-transfer resistance (R_{ct}) than that of ssDNA (Supplementary Fig. 8b).”

Fig. 2 | Reversible wettability control in a nanopipette. a, Design of a DNAzyme-based switch (DNAzyme 1 and substrate 1) for inner surface wettability control in a nanopipette (wettability-reversal nanochannel). State i: hydrophobic. State ii: Hydrophilic. **b**, Current rectification ratio measured in a wettability-reversal nanochannel upon the treatment of different concentrations of Mg²⁺ and substrate 1 (sub-1).

Supplementary Fig. 8 | Wettability characterization of DNA modified surfaces at state i and state ii. a, Current-voltage (I - V) curves measured in a wettability-reversal nanochannel before and after the treatment of Mg²⁺. **b**, Nyquist plot of DNA modified surfaces (planar glass plates modified with DNAzyme 1/Sub-1) without and with the treatment Mg²⁺ using 10 mM [Fe(CN)₆]³⁻/[Fe(CN)₆]⁴⁻ as redox mediators.

We have also added the description of the methods on Page 24, line 536-549 as following:

“Electrochemical impedance spectroscopic (EIS) measurement. EIS analysis was conducted in 10 mM [Fe(CN)₆]³⁻/[Fe(CN)₆]⁴⁻ dissolved in 1 × PBS (pH 7.4) containing 0.1 M KCl at room temperature. The experimental parameters were as follows: open circuit potential, +0.25 V; potential amplitude, ±10 mV; frequency range, 0.1-100000 Hz.”

“Contact angle measurement. Contact angle (CA) measurements were performed via an OCA 20 system to collect the CA data of planar glass plates at different states: 1. without any modification (control); 2. with the modification of DNAzyme 1/Sub-1 (without and with the treatment of Mg²⁺, state i/ii); 3. with the modification of

DNAzyme 2/Sub-2(without and with the treatment of Zn^{2+} , state iii/iv), 4. with the modification of mixture solution of DNAzyme 1/Sub-1 and DNAzyme 2/Sub-2 (state 1); 5. with the modification of mixture solution of DNAzyme 1 and DNAzyme 2/Sub-2 (state 2); 6. with the modification of mixture solution of DNAzyme 1/Sub-1 and DNAzyme 2 (state 3); 7. with the modification of mixture solution of DNAzyme 1 and DNAzyme 2 (state 4). The average CA value was obtained from three different locations.”

Comment 2. *Additionally, the result that, as shown in Fig.4, the current level of state 1 is higher than that of state 2, conflicts the hydrophobic barrier mechanism in page 5 claimed by the authors, I think that since the presented nanochannel size is much larger than the critical size, which is the diameter of an ion's first hydration shell, the hydrophobic barrier mechanism is not applicable. Rather, hydrophobic nanochannel can show high conductivity and flow compared to the hydrophilic nanochannel owing to the hydrodynamic slip[1-2].*

[1] Kun Li et al., *Anomalous ion transport through hydrophilic and hydrophobic nanopores, 2017 IEEE International Conference on Manipulation, Manufacturing and Measurement on the Nanoscale (3M-NANO), 151-155 (2017).*

[2] Olga I. Vinogradova et al, *Transport of ions in hydrophobic nanotubes, Physics of Fluids 34, 122003 (2022).*

Reply:

We thank the reviewer for the insightful comments. We are sorry for the contradictory description in page 5 in the original submission. We agree with the reviewer that the hydrophobic barrier mechanism is not applicable, since the presented nanochannel size is much larger than the critical size. As mentioned in the reply for comments 1, here in Fig. 4, the cholesterol tail was tagged at the end of sub-1. Upon hybridization between DNAzyme 1 and sub-1, the inner surface of the nanochannel was modified by the hybridized DNA initially (dsDNA) (state 1). When Mg^{2+} was added, the sub-1 strand was cleaved, and the dsDNA (DNAzyme 1/sub-1) changed to single stranded DNA (ssDNA) (state 2). As a result, the surface charge at state 1 was higher than that at state 2, which could be demonstrated by electrochemical impedance spectroscopy. From Nyquist plots (Supplementary Fig. 16a), it could be observed that state 1 showed a higher R_{ct} than state 2. Therefore, the nanochannel was in the hydrophobic surface (state 1) with a higher ICR level than that at state 2 at -1 V.

The two references provided by the reviewer reported that hydrophobic nanochannel can show high conductivity and flow compared to the hydrophilic nanochannel owing to the hydrodynamic slip. In general cases, the surface charge of hydrophobic surface is lower than hydrophilic nanochannel. Actually, in this work, the surface modification was different from the general situation, as the DNA backbones could provide negative surface charge despite of the cholesterol tail. The lipophilic cholesterol groups at the end of sub-1 can form hydrophobic interaction with FITC, a lipophilic dye used in the experiments to test the permeability of the functionalized nanochannels switchable between hydrophobic and hydrophilic states. As shown in Fig. 5a & b, the fluorescence (FL) of intracellular FITC after injection was significantly higher in cell 2' and cell 4',

indicating an increase lipid permeability in state 2 and state 4 (without the cholesterol tail). These results indicated that the cholesterol groups at the end of sub-1 could serve as absorbents to trap FITC molecules, which enabled controlled directional flow of FITC in the nanochannel. Furthermore, from static contact angle experiments (Supplementary Fig. 16b), it could be observed that the contact angle detected at the four different states (state 1, state 2, state 3 and state 4) was ($47.31 \pm 2.13^\circ$), ($26.74 \pm 1.84^\circ$), ($46.32 \pm 1.15^\circ$) and ($36.75 \pm 1.57^\circ$) respectively, which confirmed the interconvertible hydrophobicity of the nanochannels. The related descriptions have been added in Page 13, Lines 273-292 as following:

“In order to achieve more delicate control for the regulation of wettability and charge, the inner wall of the nanopipettes was modified with two pairs of double chains, DNAzyme 1/sub-1 and DNAzyme 2/sub-2, and the as-prepared artificial nanochannels were designed to be switchable between four different permeability states, including state 1 (hydrophobization, negative rectification), state 2 (hydrophilization, negative rectification), state 3 (hydrophobization, positive rectification), and state 4 (hydrophilization, positive rectification). The corresponding *I-V* curves under the four different states were tested (Fig. 4a). The ICR ratio of state 1 was 3.36, indicating a negatively charged inner surface. After Mg^{2+} was added, the ICR ratio changed from 3.36 to 1.90 due to the falling out of sub-1 with cholesterol tail, and the nanochannel was then switched from state 1 to state 2. The ICR ratios of state 3 and state 4 were further decreased to 0.87 and 0.53 respectively, which could be attributed to the exposure of amino groups upon the addition of Zn^{2+} . From Nyquist plots (Supplementary Fig. 16a), it could be observed that state 1 showed a higher R_{ct} than state 2, while state 4 represented the minimum R_{ct} value. The contact angle at different states (state 1, state 2, state 3 and state 4) was detected to be ($47.31 \pm 2.13^\circ$), ($26.74 \pm 1.84^\circ$), ($46.32 \pm 1.15^\circ$) and ($36.75 \pm 1.57^\circ$) respectively (Supplementary Fig. 16b), which confirmed the interconvertible hydrophobicity of the nanochannels, since higher contact angles indicated less wetting surfaces. The above results demonstrated that the nanochannel could be reversibly switched between these four different states.”

Fig. 4 | Rectifying artificial nanochannels with four interconvertible permeability states. **a**, I - V curves measured in an as-prepared nanochannel corresponding to state 1, state 2, state 3 and state 4. **b**, Left: Schematic illustration showing the reversal switch between state 1 and state 2. Right: Current rectification ratio measured at state 1 and state 2. **c**, Left: Schematic illustration showing the reversal switch between state 2 and state 3. Right: Current rectification ratio measured at state 2 and state 3. **d**, Left: Schematic illustration showing the reversal switch between state 3 and state 4. Right: Current rectification ratio measured at state 3 and state 4. **e**, Left: Schematic illustration showing the reversal switch between state 4 and state 1. Right: Current rectification ratio measured at state 4 and state 1. **f**, Left: Schematic illustration showing the reversal switch between state 1 and state 3. Right: Current rectification ratio measured at state 1 and state 3. **g**, Left: Schematic illustration showing the reversal switch between state 2 and state 4. Right: Current rectification ratio measured at state 2 and state 4.

Fig. 5 | Controlled Directional Flow in Artificial Nanochannels (displayed a and b pannels). Microscopic cell images showing the outflow of FITC (**a, b**) into single living HeLa cells before and after injection. BF: bight-field. FITC: green fluorecence. Merge: mixed green & bright-field channel.

Supplementary Fig. 16 | Nyquist plots and contact angles of state 1, 2, 3 and 4. **a**, Nyquist plots of DNA modified surfaces of state 1, 2, 3 and 4 using $1 \times [\text{Fe}(\text{CN})_6]^{3-} / [\text{Fe}(\text{CN})_6]^{4-}$ as redox mediators. **b**, Photographs of the water droplets on DNA modified surfaces of state 1, 2, 3 and 4. The average contact angle value was obtained from the value detected at three different locations.

Comment 3. *Regarding this issue, the authors should suggest another proof way for hydrophobicity of nanochannel beside of change of rectification or clear explanation for these unexpected results.*

Reply:

We appreciate the reviewer's comments and have followed the advice by performing static contact angle experiments to reflect that the wettability of nanochannels at different states (state i, ii, iii, iv and state 1, 2, 3, 4). As shown in Supplementary Fig. 8c, the contact angle decreased from ($46.43 \pm 1.21^\circ$) to ($26.39 \pm 1.97^\circ$) after the addition of Mg^{2+} , indicating that the cleavage of cholesterol tagged sub-1 induced a more hydrophilic surface at state ii than that at state i. Thus, the addition of Mg^{2+} transformed the surface wettability from hydrophobic to hydrophilic.

The related descriptions have been added in Page 10, Lines 201-205, Page 12, Lines 243-245 and Page 13, Lines 287-292 as following:

“In addition, static contact angle experiments showed that the contact angle decreased from ($46.43 \pm 1.21^\circ$) to ($26.39 \pm 1.97^\circ$) after the addition of Mg^{2+} (Supplementary Fig. 8c), indicating that the cleavage of cholesterol tagged sub-1 induced a more hydrophilic surface at state ii than that at state i. Thus, the addition of Mg^{2+} transformed the surface wettability from hydrophobic to hydrophilic”

Supplementary Fig. 8 | Wettability characterization of DNA modified surfaces at state i and state ii. **c**, Photographs of the water droplets on DNA modified surfaces (planar glass plates modified with DNAzyme 1/Sub-1) without and with the treatment of Mg^{2+} . The average contact angle value was obtained from the value detected at three different locations.

“Static contact angle experiments showed similar contact angles of state iii ($27.27 \pm 0.98^\circ$) and state iv ($24.52 \pm 0.65^\circ$) (Supplementary Fig. 12c).”

Supplementary Fig. 12 | Wettability characterization of DNA modified surfaces at state iii and state iv. **c**, Photographs of the water droplets on DNA modified surfaces (planar glass plates modified with DNAzyme 2/Sub-2) without and with the treatment of Zn^{2+} . The average contact angle value was obtained from the value detected at three different locations.

“The contact angle at different states (state 1, state 2, state 3 and state 4) was detected to be ($47.31 \pm 2.13^\circ$), ($26.74 \pm 1.84^\circ$), ($46.32 \pm 1.15^\circ$) and ($36.75 \pm 1.57^\circ$) respectively (Supplementary Fig. 16b), which confirmed the interconvertible hydrophobicity of the nanochannels. The above results demonstrated that the nanochannel could be reversibly switched between these four different states.”

Supplementary Fig. 16 | Nyquist plots and contact angles of state 1, 2, 3 and 4. **b**, Photographs of the water droplets on DNA modified surfaces of state 1, 2, 3 and 4. The average contact angle value was obtained from the value detected at three different locations.

Comment 4. *Rectification of tapered nanochannel comes from the EDL overlap. So, highly concentrated medium can induce disadvantage for rectification. The medium in this study is PBS having high concentration of NaCl. The authors should check the theoretical Debye layer thickness is reasonable considering the geometrical size of nanochannel for the proper operation in this medium.*

Reply:

We agree with the reviewer that highly concentrated medium could induce disadvantage for rectification. We are sorry for not being clear in describing the concentration of the PBS buffer. In this work, the electrolyte solution was diluted PBS ($0.1 \times$ PBS containing 1 mM PO_4^{3-} , 13.7 mM NaCl, 0.27 mM KCl). The theoretical Debye layer thickness is calculated to be 3 nm, which is reasonable according to

previous researches.^{Ref. R1-5} The related descriptions have been changed in Page 24, Lines 526-529 as:

“Two Ag/AgCl wires were introduced for obtaining electrochemical signals, as one of them was used as the working electrode inserted into the nanochannel filled with electrolyte solution ($0.1 \times$ PBS, pH 7.4), and the other one was used as the reference electrode bathed in DMEM solution.”

[Ref. R1]. Xiong, T.Y. et al. Neuromorphic functions with a polyelectrolyte-confined fluidic memristor. *Science* **379**, 156-161 (2023).

[Ref. R2]. Song, J. et al. Ultrasmall nanopipette: toward continuous monitoring of redox metabolism at subcellular level. *Angew. Chem. Int. Ed.* **57**, 13226-13230 (2018).

[Ref. R3]. Jia, R., Mirkin, M. V. The double life of conductive nanopipette: a nanopore and an electrochemical nanosensor. *Chem. Sci.* **11**, 9056-9066 (2020).

[Ref. R4]. Pan, R. R., Wang, D. C., Liu, K., Chen, H. Y. & Jiang, D. C. Electrochemical molecule trap-based sensing of low-abundance enzymes in one living cell. *J. Am. Chem. Soc.* **144**, 17558-17566 (2022).

[Ref. R5]. Ding, H., Liu, K., Zhao, X. L., Su, B. & Jiang, D. C. Thermoelectric nanofluidics probing thermal heterogeneity inside single cells. *J. Am. Chem. Soc.* (2023).

Comment 5. *As far as I know, MB has no red fluorescent excitation (Maybe I wrong). Is there any fluorescent tagging for MB in this study? If not, can the author add the explanation about the origin of red fluorescent signal in MB?*

Reply:

According to the reviewer's suggestion, we have performed additional experiments to test the fluorescent properties of MB by ultraviolet (UV) spectra, fluorescence (FL) spectra and confocal microscopic imaging. As shown in Supplementary Fig. 15a, the UV spectra of MB exhibited an absorption peak at 660 nm, which was in accordance with the product description of MB (Response Fig. R1). The FL spectra of MB showed an emission peak at 680 nm, indicating red fluorescence of MB (Supplementary Fig. 15b). The confocal microscopic image of MB incubated with HeLa cells also showed red fluorescence under 561 nm excitation (Supplementary Fig. 15c).

The related description has been added in Page 12, Line 262-265 as following:

“To mimic controlled directional flow in natural transmembrane channels, methylene blue (MB, a commonly used cationic dye for biological staining with red fluorescence, Supplementary Fig. 15) was selected to test the permeability of the functionalized nanochannel (Fig. 3d).”

Supplementary Fig. 15 | UV-vis, FL characterizations and confocal microscopic images of MB dyes. a, UV-vis spectra of MB dyes. **b,** FL spectra of MB dyes. **c,** Microscopic cell images of living HeLa cells incubated with MB dyes. The cells were excited at 561 nm, and the emissions were collected from 662 nm-737 nm. BF: bright-field. MB: red fluorescence. Merge: mixed red & bright-field channel. Scale bar: 20 μm .

Name Methylene blue Solution, 100X 1 Solution
Cat. No. A610622
Product No. A610622-0025
CAS# [7220-79-3]
Formula $\text{C}_{16}\text{H}_{18}\text{ClN}_3\text{S}\cdot 3\text{H}_2\text{O}$
Lot. No. J718BA0019

VERSION #Q0001309_20230718

TEST	SPECIFICATION	ANALYSIS
Appearance	Dark blue liquid	PASS
Maximum Absorption Wavelength(λ_{max} , H ₂ O)	662.0nm~668.0nm	664.4nm
Biological Staining Test	Confirmed	PASS

Fig. R1 | The product description of MB.

Fig. 3 | Reversible charge control in a nanopipette. d, Schematic illustration showing transport of MB in a charge-reversal nanochannel under a DC voltage into single living HeLa cells.

Response to reviewer 3:

General Comment:

Drawing inspiration from transmembrane ion channels, the authors have engineered artificial nanochannels that mimic the selective permeation function of these biological channels. These artificial constructs emerge as promising tools with a multitude of potential applications in the realm of biomedical research.

In this study, the authors delineate a strategy for fabricating biomimetic nanochannels using glass nanopipettes, internally modified with DNAzymes capable of cleaving RNA. DNAzymes are specific DNA sequences exhibiting enzymatic activity that necessitate a particular metal ion cofactor for their function. Through this innovation, the authors successfully achieved reversible switching of the nanopipette's permeability across four distinct states. Moreover, upon integration of these nanochannels into living cells, they demonstrated selective transport of bioactive molecules.

To institute the four permeability states within the nanopipettes, their internal wall was modified using two specific DNAzymes: one targeting Mg^{2+} and the other Zn^{2+} . With these DNAzymes anchored, it became possible to modulate the nanochannel's permeability across four configurations: hydrophobic with negative rectification, hydrophilic with negative rectification, hydrophobic with positive rectification, and hydrophilic with positive rectification.

Two methodologies were postulated for the internal modification of the pipettes: the induction of aldehyde groups and a gold coating. After evaluating the stability of both approaches, it was ascertained that the bond between the -CHO and -NH₂ groups was more robust, leading to the selection of this methodology for subsequent investigations. Once these implementations have been achieved, the authors carry out a series of exciting demonstrations that allow evaluating Reversible wettability control in a nanopipette, Reversible charge control in a nanopipette, Rectifying artificial nanochannels with four interconvertible permeability states. The Evaluation and simulation of nanopipette-based transmembrane delivery. All these demonstrations are done using different electrophysiological tools that are well-documented and validated in the article.

Although the article is very well implemented experimentally, I note that the main weakness of the article is conceptual. In the section titled "Design of artificial nanochannels based on DNAzyme-functionalized nanopipettes," the description of biological channels is too generalized, and the channel concept is forced to emphasize the similarity of these biological entities with functionalized nanopipettes. Specifically, the pores of the channels are not all hydrophobic. Furthermore, the description of the selectivity filter differs from the reality of many known ion channels. For example, the K channels have particular selectivity filters, constituted by the backbone of a sequence of uncharged residues, TVGYG. Therefore, I request that the authors provide a more detailed and precise description of the biological entity that they used as a reference, which could be a specific type of ion channel or another transmembrane protein that better reflects the findings of the characteristics presented in this study.

Reply:

We appreciate the reviewer's positive comments and agree with the reviewer that the descriptions about biological channels and the selectivity filter need to be improved. We followed the reviewer's advice by rewriting the related part in the section "Design of artificial nanochannels based on DNAzyme-functionalized nanopipettes", as a more detailed and precise description of the biological entity that better reflects the findings of the characteristics presented in this study has been discussed in Page 5, Lines 103-111 as following:

"Biological channels share a common feature of containing narrow pore channels with selective filter regions to achieve fast and selective transport of various bioactive species in and out of cells.⁴⁰ The selective transport mechanism of biological channels has been considered as an efficient regulator to control the permeation of ions and molecules.⁴¹ In some particular conditions, the inner surface property of the channels can be reversed to allow the passing through of certain ions or molecules. A typical example is the Orai calcium channels. As shown in Fig. 1a, in the closed state of Orai channel, the permeability of Ca^{2+} is blocked. Upon activation, the exposure of alkaline amino acid rich region causes strong attraction of anions, which enables Ca^{2+} permeation.⁴²"

Fig. 1 | a, Orai channels with Ca^{2+} selective permeability in nature.

References:

40. Lin, X. Y., Yang, Q., Yan, F., Zhang, B.W. & Su, B. Gated molecular transport in highly ordered heterogeneous nanochannel array electrode. *ACS Appl. Mater. Interfaces*, **8**, 33343-33349, (2016).
41. Beckstein, O. & Sansom, M. S. P. The influence of geometry, surface character, and flexibility on the permeation of ions and water through biological pores. *Phys. Biol.* **1**, 42 (2004)
42. Liu, X. F. et al. Molecular understanding of calcium permeation through the open Orai channel. *PLOS Biol* **17**, e3000096 (2019).

REVIEWER COMMENTS

Reviewer #1 (Remarks to the Author):

This revised manuscript is much better than original version, and all my concerns have been addressed carefully, I think this manuscript can be accepted for publication in Nature Communications as its present form.

Reviewer #2 (Remarks to the Author):

After reviewing the authors' response to the comment, I found that the authors tried their best to address the comment by carrying out additional experiments and analysis and adding explanations to resolve the queries. Overall, the revised version and additional supplemental data made up for the previous issues raised by the reviewers well.

However, I still have one issue that should be cleared before publication.

The authors claimed that the hydrophobic state (I) has more negative surface charge compared to the hydrophilic state (II), and the main ground is the Nyquist plot for Nyquist plots for DNA-modified surfaces before and after the addition of Mg²⁺: The higher charge transfer resistance (R_{ct}) at state (I) means a more negative surface charge at state (I). As far as I know, the charge transfer resistance shows that there is some dielectric layer (from biological or other synthetic materials) at the electrode that prohibits charge transfer from the surface. Therefore, I think the higher charge transfer reflects the conformal formation of the DNA-modified surface after the addition of Mg²⁺.

I'm curious whether there is a direct link between the charge transfer resistance and the surface charge. In my opinion, to prove such a high surface charge, I think the measurement of the zeta potential of the modified surface is a direct method. Perhaps if there is a strong link between charge transfer resistance and negative surface charge, please explain it clearly.

Reviewer #3 (Remarks to the Author):

The revised version of the article fully adheres to the suggested changes. Therefore, no further comments are necessary.

Response Reviewers' Comments

We sincerely thank the reviewers for their valuable comments and constructive questions. According to the comments and editor's suggestion, we have revised the manuscript carefully. Here are our responses to reviewers' comments and changes we have made. All the changes are highlighted by yellow background in the revised manuscript.

Response to reviewer 1:

General Comment:

This revised manuscript is much better than original version, and all my concerns have been addressed carefully, I think this manuscript can be accepted for publication in Nature Communications as its present form.

Reply:

We appreciate the reviewer's positive comments and valuable suggestions.

Response to reviewer 2:

General Comment:

After reviewing the authors' response to the comment, I found that the authors tried their best to address the comment by carrying out additional experiments and analysis and adding explanations to resolve the queries. Overall, the revised version and additional supplemental data made up for the previous issues raised by the reviewers well. However, I still have one issue that should be cleared before publication. The authors claimed that the hydrophobic state (I) has more negative surface charge compared to the hydrophilic state (II), and the main ground is the Nyquist plots for DNA-modified surfaces before and after the addition of Mg^{2+} : The higher charge transfer resistance (R_{ct}) at state (I) means a more negative surface charge at state (I). As far as I know, the charge transfer resistance shows that there is some dielectric layer (from biological or other synthetic materials) at the electrode that prohibits charge transfer from the surface. Therefore, I think the higher charge transfer reflects the conformal formation of the DNA-modified surface after the addition of Mg^{2+} .

I'm curious whether there is a direct link between the charge transfer resistance and the surface charge. In my opinion, to prove such a high surface charge, I think the measurement of the zeta potential of the modified surface is a direct method. Perhaps if there is a strong link between charge transfer resistance and negative surface charge, please explain it clearly.

Reply:

We appreciate the reviewer's positive comments and valuable suggestions. We agree with the reviewer that the higher charge transfer reflects the conformal formation of the DNA-modified surface after the addition of Mg^{2+} . In this work, the electrochemical impedance spectroscopic (EIS) analysis was conducted in 10 mM $[Fe(CN)_6]^{3-}/[Fe(CN)_6]^{4-}$ dissolved in $1 \times$ PBS (pH 7.4). According to the principle of like charges repel, the negatively charged surface of the ITO electrode could prevent the

approaching of $[\text{Fe}(\text{CN})_6]^{3-}/[\text{Fe}(\text{CN})_6]^{4-}$. For state i, the electrode surface was modified by the hybridized DNA (dsDNA) upon the hybridization between DNAzyme 1 (a Mg^{2+} -specific DNAzyme) and its substrate (sub-1). After the addition of Mg^{2+} , the sub-1 strand was cleaved, and the electrode surface was then switched from state i to state ii, with the surface modification changed to single-stranded DNA (ssDNA). Due to the negative charge of DNA backbone, the conformation of the dsDNA exposed the negatively charged backbone to the surface more fully than the ssDNA, the negatively charged $[\text{Fe}(\text{CN})_6]^{3-}/[\text{Fe}(\text{CN})_6]^{4-}$ would face greater resistance in approaching to the dsDNA modified surface (state i) than to the ssDNA modified surface (state ii). Therefore, state i showed a higher charge-transfer resistance (R_{ct}) compared with state ii (Supplementary Fig. 8b).⁴⁸

We also agree with the reviewer that the measurement of the zeta potential of the modified surface is a direct method. Therefore, here the measurement of the zeta potential of the modified surface before and after the treatment of Mg^{2+} was supplemented. As shown in Supplementary Fig. 8b, the zeta potential of the dsDNA modified surface (state i) was -58.93 mV, which was more negative than that of state ii (-49.89 mV), indicating a more negative surface charge at state i.

The related descriptions have been added in Page 10, Lines 196-207 as following:

“As shown in Fig. 2b, the addition of Mg^{2+} led to the cleavage of cholesterol tagged sub-1, thus the inner surface of the nanochannel changed from state i containing dsDNA to state ii containing ssDNA. Due to the negative charge of DNA backbone, the conformation of the dsDNA exposed the negatively charged backbone to the surface more fully than the ssDNA, the negatively charged $[\text{Fe}(\text{CN})_6]^{3-}/[\text{Fe}(\text{CN})_6]^{4-}$ would face greater resistance in approaching to the dsDNA modified surface (state i) than to the ssDNA modified surface (state ii). Therefore, state i showed a higher charge-transfer resistance (R_{ct}) compared with state ii (Supplementary Fig. 8b).⁴⁸ In addition, the zeta potential of the dsDNA modified surface (state i) was -58.93 mV, which was more negative than that of state ii (-49.89 mV), indicating a more negative surface charge at state i (Supplementary Fig. 8b).”

Fig. 2 | Reversible wettability control in a nanopipette. b, Current rectification ratio measured in a wettability-reversal nanochannel upon the treatment of different concentrations of Mg^{2+} and substrate 1 (sub-1).

Supplementary Fig. 8 | Wettability characterization of DNA modified surfaces at state i and state ii. **b**, Nyquist plot of DNA modified surfaces (planar glass plates modified with DNAzyme 1/Sub-1) without and with the treatment Mg^{2+} using 10 mM $[Fe(CN)_6]^{3-}/[Fe(CN)_6]^{4-}$ as redox mediators. The inset table: the zeta potential values of DNA modified surfaces (planar glass plates modified with DNAzyme 1/Sub-1) without and with the treatment of Mg^{2+} .

Reference:

48. Li, Z. L., Lv, Y. G., Duan, X. M., Liu, B. W. & Zhao, Y. X. Highly uniform DNA monolayers generated by freezing-directed assembly on gold surfaces enable robust electrochemical sensing in whole blood. *Angew Chem Int Ed Engl* **62**, e202312975 (2023).

Response to reviewer 3:

General Comment:

The revised version of the article fully adheres to the suggested changes. Therefore, no further comments are necessary.

Reply:

We appreciate the reviewer's positive comments and valuable suggestions.

REVIEWERS' COMMENTS

Reviewer #2 (Remarks to the Author):

The authors addressed the issue faithfully and well. I have no additional comments.